# A General Feature Attribution Framework under a Black-box Setting

## Abstract

Feature attribution is widely accepted as a form of explanation for reasoning machine decisions, indicating the proportion of each feature's contribution to an inquired decision. While most efforts have focused on determining attributions through exact gradient measurements, recent work has adopted gradient estimation to derive explanatory information requiring only query-level access – a restricted yet more practical accessibility assumption known as the black-box setting. Following this direction, this paper extends the idea of utilizing estimated gradients to a broader framework and introduces GEFA ((Gradient-estimation-based Explanation For All)). Unlike the previous attempt that focused on explaining image classifiers, the proposed explainer derives feature attributions in a proxy space, making it generally applicable to arbitrary black-box models, regardless of input type. In addition to its close relationship with Integrated Gradients, we find, surprisingly, that our approach – a path method built upon estimated gradients – outputs unbiased estimates of Shapley Values. By avoiding the potential information waste sourced from computing marginal contributions, it improves the quality of derived explanations, as demonstrated by our quantitative evaluations.

## 1 Introduction

With the explosive growth of deep learning models, explainability has become an increasingly important research topic. While data-driven models excel in performance, their opaque nature, originating from the implicit learning processes, raises concerns and risks, particularly when deployed in critical domains such as medical diagnosis, finance, and autonomous driving. The demand for transparency has seen the development of various techniques, including feature attribution, which is the focus of this work.

Current attempts to determine feature attribution typically fall into two categories depending on the model accessibility assumption: the white-box and black-box methods. White-box approaches assume full access to a model, deriving explanations by investigating in detail the model's internal workings through, for example, analysis of gradients (Simonyan et al., 2014; Sundararajan et al., 2017) or supervision of information flow (Samek et al., 2021). Albeit beneficial to explanation procedures, the full accessibility assumption limits the applicability of white-box approaches under practical settings due to safety and security concerns. Models deployed for public use are usually wrapped by limited APIs and accessible only via queries. On the other hand, the black-box explainers, following the assumption of query-level access, determine feature attributions by analyzing the correlation between input features and model outcomes (Ribeiro et al., 2016). As a trade-off for the loosened accessibility assumption, black-box explanations tend to be less precise, especially when explaining models operating in high-dimension feature spaces. This is because inferring explanatory information indirectly from queries is computationally expensive, with the cost positively correlated to the dimensionality of the feature space.

To combine the strengths of both categories, Cai & Wunder (2024) proposes GEEX, a path method built upon gradient estimation. Focused on the problem of explaining image classifiers, GEEX delivers gradient-like explanations under a black-box setting, achieving a performance that matches white-box explainers. However, the discussion made is limited to models that take continuous features as inputs, and the method struggles with discrete or categorical features like texts. This limitation arises from GEEX's reliance on path integral, which is not well-defined in discrete feature spaces.

While applying GEEX at the embedding layer is indeed a reasonable circumvention, it is arguable that transforming from the original feature space to some embedding space already accesses internal model details, thereby violating the black box assumption.

Bridging the gap in the applicability to models operating on discrete data, this paper extends the idea of gradient-estimation-based explanation and introduces GEFA[1] (Gradient-estimation-based Explanation For All), a general feature attribution framework built upon carefully designed proxy variables. These proxy variables facilitate the implementation of gradient estimation and path integral, regardless of input types or formats. The proposed method comes with strong theoretical guarantees. First, GEFA is an unbiased calculator of Shapley Values (Shapley, 1953), which is demonstrated through rigorous mathematical proof. Compared to previous attempts in computing Shapley Values, GEFA reduces potential information waste in sampling-based estimations, which compute marginal contributions Mitchell et al. (2022), and avoids calculations of factorials in the kernel method (Lundberg & Lee, 2017) for determining sample weights. Second, we show that our black-box explainer differs from Integrated Gradients (IG), a white-box approach by (Sundararajan et al., 2017), in only the path choice. It is proved that the two approaches become equivalent when their paths are aligned, emphasizing the connection between the gradient-estimation-based approach and actual gradients. Finally, we design a simple control variate that is guaranteed to improve explanation quality under a simple and realistic assumption. Its effectiveness is demonstrated through quantitative experiments across various settings.

## 2 RELATED WORK

Gradients are widely used to allocate feature attributions in a white-box setting as they reveal a model's sensitivity to changes in feature values. In the early development of explainability, Simonyan et al. (2014) and Smilkov et al. (2017) interpreted gradients directly as explanations. Their methods retrieve explanatory information by tracing partial derivatives of a decision function with respect to its input features. Although adopting vanilla gradients is a reasonable starting point, gradients by themselves reflect local sensitivity and do not truthfully represent contributions of feature presence without a proper definition of feature absence.

IG (Sundararajan et al., 2017) addresses the limitation of vanilla gradients with a baseline point modeling feature absence. The approach integrates gradients over a straightline path connecting the baseline and the explaining target, thereby capturing the overall impact of feature presence. Following work by Sturmfels et al. (2020) explored the impact of baseline choice and suggested adopting a distribution, rather than a deterministic instance, as the baseline (Erion et al., 2021). Other extensions of IG include decomposing noise directions from the path integral (Yang et al., 2023), refining explanations by filtering out high frequencies (Muzellec et al., 2024), and investigating feature interactions through the integration of second-order derivatives (Janizek et al., 2021). Parallel to these efforts in improving the explanation procedure, Decker et al. (2024) demonstrated that a proper linear composition of explanations from various approaches yields provable improvements. The family of propagation-based methods (Montavon, 2019) represents a significant alternative white-box solution, which designs layer-wise back-propagation rules that explicitly utilize model architecture information for the retrieval of explanatory information. As this paper focuses primarily on gradient-based and gradient-like explanations, we refer interested readers to the survey by (Samek et al., 2021) for further details on relevance propagation.

Unlike white-box methods, which have direct access to model details, black-box explainers determine feature attributions by collecting and analyzing observations. The idea was proposed by LIME (Ribeiro et al., 2016), which generates queries by altering feature values of the original input and collects model responses to the perturbed instances. By solving a linear regression problem with the observed input-output pairs, LIME derives regressor coefficients as feature attributions. Subsequently, Lundberg & Lee (2017) proposed KernelSHAP, a kernel method that approximates Shapley Values using weighted linear regression. Additionally, Lundberg & Lee (2017) formalized the relationship between the feature attribution problem and cooperative game theory, strengthening the importance of Shapley Values in explainability.

---

[1]Code for reproducibility can be found at: https://hide.for.anonymity

Under the established framework of black-box approaches, succeeding works have aimed at improving query efficiency and explanation quality – long-standing challenges for black-box approaches. For example, Dhurandhar et al. (2022) extended LIME with an adaptive neighborhood sampling scheme that constrains sampling to locally linear regions based on the explicand. Petsiuk et al. (2018) alleviated concerns about computational expenses by softly grouping input features via mask resizing, effectively reducing the dimensionality of the feature space. Similarly, Shrotri et al. (2022) and Dhurandhar et al. (2024) improved sampling efficiency by narrowing the search space. Parallel to refining the sampling process, Frye et al. (2020) and Heskes et al. (2020) enhanced explanation quality by incorporating prior causal knowledge into the SHAP framework. Okhrati & Lipani (2021) leveraged the multilinear extension method from game theory literature (Owen, 1972) to develop a sampling-based explainer with reduced variance. More recently, Cai & Wunder (2024) adopted gradient estimation and imitated IG under a black-box setting by integrating estimated gradients, resulting in white-box-level performance with query-level access. However, this approach is limited to continuous feature space, which is the gap addressed in the following sections.

## 3 PRELIMINARY

### 3.1 FEATURE ATTRIBUTION

Given a model function $f(\cdot)$, a target input (the explicand) $\boldsymbol{x} = (x_1, x_2, \ldots, x_p)$, and a predefined baseline $\mathring{\boldsymbol{x}} = (\mathring{x}_1, \mathring{x}_2, \ldots, \mathring{x}_p)$, an attribution method seeks a vector $\boldsymbol{\xi} \in \mathbb{R}^p$ that decomposes the total contribution to an inquired decision into feature attributions. Formally, this is represented as:

$$A_f : (\boldsymbol{x}, \mathring{\boldsymbol{x}}) \hookrightarrow (\xi_1, \xi_2, \ldots, \xi_p)$$

Throughout the paper, we mark vectors in bold and denote scalars with plain symbols.

As a result of allocating feature contributions, the attribution scores $\xi_i$ indicate the contribution of each feature $x_i$ to the model outcome $f(\boldsymbol{x})$, and they should sum up to the difference between the model outcome with all features present and the outcome with full feature absence, which is modeled by the baseline:

$$\sum_{i=0}^{p} \xi_i = f(\boldsymbol{x}) - f(\mathring{\boldsymbol{x}}) \tag{1}$$

Approaches complying with equation 1 are said to satisfy the property of *Completeness* – a fundamental aspect of feature attribution methods. Together with completeness, further properties are desired for feature attribution methods, which upholds the practical meanings of feature attribution:

- *Sensitivity*: a feature should receive non-zero attribution if the difference of its value between the explicand and the baseline induces a change in model outcomes

- *Insensitivity*: the attribution should be zero for any feature, on which the model is functionally independent

- *Linearity*: the explanation for the linear composition of two functions should equal the weighted sum of the separate explanations for them

- *Symmetry*: if a function is symmetric in two variables $x_i$ and $x_j$, the attributions to the two features should be the same when the explicand-baseline pair holds $x_i = x_j$ and $\mathring{x}_i = \mathring{x}_j$

### 3.2 GRADIENT ESTIMATION UNDER A BLACK-BOX SETTING

In the context of feature attribution, a black box setting refers to query-level access, meaning that the model to be explained can only be accessed via its input and output interfaces. Indeed, lacking knowledge about the model's internal details prohibits the application of attribution methods that depend on exact measurements of gradients. However, gradients, which facilitate the derivation of feature attributions, can still be estimated by evaluating model inputs and outputs. Defining a search distribution $\boldsymbol{\pi}(\cdot|\boldsymbol{x})$ parameterized by $\boldsymbol{x}$, the expected model outcome over $\boldsymbol{\pi}(\cdot|\boldsymbol{x})$ is given by:

$$J(\boldsymbol{x}) := \mathbb{E}_{\boldsymbol{\pi}(\boldsymbol{z}|\boldsymbol{x})}[f(\boldsymbol{z})] = \int f(\boldsymbol{z})\boldsymbol{\pi}(\boldsymbol{z}|\boldsymbol{x})\,\mathrm{d}\boldsymbol{z} \tag{2}$$

where $z$ indicates samples drawn from the search distribution. The gradient of the expected model outcome with respect to $x$ is:

$$\nabla_{\boldsymbol{x}} J(\boldsymbol{x}) = \nabla_{\boldsymbol{x}} \int f(\boldsymbol{z})\boldsymbol{\pi}(\boldsymbol{z}|\boldsymbol{x})\,\mathrm{d}\boldsymbol{z} \tag{3}$$

The above formula can be further simplified using the log-likelihood trick, under the assumption that both $f(\cdot)$ and $\boldsymbol{\pi}(\cdot|\boldsymbol{x})$ are continuously differentiable (Mohamed et al., 2020):

$$\nabla_{\boldsymbol{x}} J(\boldsymbol{x}) = \int [f(\boldsymbol{z}) \cdot \nabla_{\boldsymbol{x}} \log \boldsymbol{\pi}(\boldsymbol{z}|\boldsymbol{x})]\boldsymbol{\pi}(\boldsymbol{z}|\boldsymbol{x})\,\mathrm{d}\boldsymbol{z}$$
$$= \mathbb{E}_{\boldsymbol{\pi}(\boldsymbol{z}|\boldsymbol{x})}[f(\boldsymbol{z}) \cdot \nabla_{\boldsymbol{x}} \log \boldsymbol{\pi}(\boldsymbol{z}|\boldsymbol{x})] \tag{4}$$

The integral can be empirically approximated with a Monte Carlo estimator with a set of queries $\boldsymbol{Z} = \{\boldsymbol{z}|\boldsymbol{z} \sim \boldsymbol{\pi}(\cdot|\boldsymbol{x})\}$, leading to the typical score-function gradient estimator:

$$\boldsymbol{\eta_x}(\boldsymbol{x}) := \nabla_{\boldsymbol{x}} J(\boldsymbol{x}) \approx \frac{1}{|\boldsymbol{Z}|} \sum_{\boldsymbol{z} \in \boldsymbol{Z}} f(\boldsymbol{z}) \cdot \nabla_{\boldsymbol{x}} \log \boldsymbol{\pi}(\boldsymbol{z}|\boldsymbol{x})$$

## 4 GRADIENT-ESTIMATION-BASED EXPLANATION FOR ALL

### 4.1 GRADIENT ESTIMATION WITH PROXY VARIABLES

Given the diverse nature of potential input features, sampling instances by perturbing feature values is not always straightforward. Instead of altering feature values by applying noises, we define the search distribution through a set of proxy variables $\boldsymbol{\alpha} = (\alpha_1, \alpha_2, \dots, \alpha_p)$. The proxy vector $\boldsymbol{\alpha}$ shares the same size as the explicand, where each element $\alpha_i$ configures the presence probability of the corresponding explicand feature $x_i$. Recalling that feature presence and absence are modeled by feature values of the explicand and the baseline, respectively. A point $\boldsymbol{x}(\boldsymbol{\alpha})$ in the continuous proxy space of $\boldsymbol{\alpha} \in [0,1]^p$ describes a distribution, with each sample $\boldsymbol{z} \sim \boldsymbol{x}(\boldsymbol{\alpha})$ is given by:

$$z_i = \begin{cases} x_i & \text{if } \epsilon_i = 1 \\ \mathring{x}_i & \text{if } \epsilon_i = 0 \end{cases} \quad \forall i \in \{1, 2, \dots, p\}$$

where $\boldsymbol{\epsilon} = (\epsilon_1, \epsilon_2, \dots, \epsilon_p)$ denotes a binary mask $\boldsymbol{\epsilon} \sim \text{Bernoulli}(\boldsymbol{\alpha})$ sampled from a multivariate Bernoulli distribution parameterized by $\boldsymbol{\alpha}$. For ease of notation, we denote the feature selection process with a feature-wise combination operator $\oplus$, which indicates a feature $z_i$ in the sample $\boldsymbol{z}$ takes the value of $\boldsymbol{x}$ when the corresponding mask component $\epsilon_i = 1$, otherwise set to $\mathring{x}_i$:

$$\boldsymbol{z} = \boldsymbol{\epsilon} \circ \boldsymbol{x} \oplus \bar{\boldsymbol{\epsilon}} \circ \mathring{\boldsymbol{x}}, \quad \boldsymbol{\epsilon} \sim \text{Bernoulli}(\boldsymbol{\alpha})$$

The vector $\bar{\boldsymbol{\epsilon}} = \mathbb{1}_p - \boldsymbol{\epsilon}$ is the complement of $\boldsymbol{\epsilon}$. The operator $\circ$ indicates the element-wise product, where a feature value is selected if the mask component equals one, otherwise it remains undefined until a value is assigned through the $\oplus$ operator. Please note that the feature selection operator does not depend on feature types and is generally applicable as long as the explicand-baseline pair is specified. Given an explicand-baseline pair, the sampling of a query $\boldsymbol{z}$ depends fully on the binary mask $\boldsymbol{\epsilon}$, whose probability mass function is:

$$\boldsymbol{\pi}(\boldsymbol{z}|\boldsymbol{x}(\boldsymbol{\alpha})) = \boldsymbol{\pi}(\boldsymbol{\epsilon}|\boldsymbol{\alpha}) = \boldsymbol{\alpha}^{\boldsymbol{\epsilon}} \cdot (\mathbb{1}_p - \boldsymbol{\alpha})^{\bar{\boldsymbol{\epsilon}}} \tag{5}$$

Here, $\boldsymbol{\alpha}^{\boldsymbol{\epsilon}}$ is a shorthand for $(\alpha_1{}^{\epsilon_1}, \alpha_2{}^{\epsilon_2}, \dots, \alpha_p{}^{\epsilon_p})$. Substituting the distribution given by equation 5 for the search distribution $\boldsymbol{\pi}$ in equation 4 yields an estimator for the gradient of $f(\boldsymbol{x}(\boldsymbol{\alpha}))$ w.r.t. the proxy parameters $\boldsymbol{\alpha}$:

$$\boldsymbol{\eta_\alpha}(\boldsymbol{x}(\boldsymbol{\alpha})) = \mathbb{E}_{\boldsymbol{\pi}(\boldsymbol{z}|\boldsymbol{x}(\boldsymbol{\alpha}))}[f(\boldsymbol{z}) \cdot \nabla_{\boldsymbol{\alpha}} \log \boldsymbol{\pi}(\boldsymbol{z}|\boldsymbol{x}(\boldsymbol{\alpha}))]$$
$$= \mathbb{E}_{\boldsymbol{\pi}(\boldsymbol{\epsilon}|\boldsymbol{\alpha})}[f(\boldsymbol{\epsilon} \circ \boldsymbol{x} \oplus \bar{\boldsymbol{\epsilon}} \circ \mathring{\boldsymbol{x}}) \cdot \nabla_{\boldsymbol{\alpha}} \log(\boldsymbol{\alpha}^{\boldsymbol{\epsilon}} \cdot (\mathbb{1}_p - \boldsymbol{\alpha})^{\bar{\boldsymbol{\epsilon}}})]$$
$$= \mathbb{E}_{\boldsymbol{\pi}(\boldsymbol{\epsilon}|\boldsymbol{\alpha})}[f(\boldsymbol{\epsilon} \circ \boldsymbol{x} \oplus \bar{\boldsymbol{\epsilon}} \circ \mathring{\boldsymbol{x}}) \cdot (\frac{\boldsymbol{\epsilon}}{\boldsymbol{\alpha}} - \frac{\bar{\boldsymbol{\epsilon}}}{\mathbb{1}_p - \boldsymbol{\alpha}})] \tag{6}$$

When referring to the logarithm of the probability vector $\boldsymbol{\pi}$, we specifically mean applying the logarithm operation element-wise to each vector component. Given that $\boldsymbol{\alpha}$ represents the probabilities of feature presence, the output of $\boldsymbol{\eta_\alpha}(\boldsymbol{x}(\boldsymbol{\alpha}))$ can be interpreted as the sensitivity of model outcomes to changes in feature presence.

## 4.2 Derivation of GEFA

In addition to promoting the derivation of the gradient estimator, the introduction of proxy parameters facilitates the integration of inputs with discrete features (e.g. text) when deriving feature attribution. Formally, let $\boldsymbol{\alpha}(\cdot) = (\alpha_1, \ldots, \alpha_p) : [0,1] \to [0,1]^p$ be a path in the proxy space from the baseline $\boldsymbol{x}(\boldsymbol{\alpha}(0)) = \boldsymbol{x}(\mathbb{0}_p) = \mathring{\boldsymbol{x}}$ to the explicand $\boldsymbol{x}(\boldsymbol{\alpha}(1)) = \boldsymbol{x}(\mathbb{1}_p) = \boldsymbol{x}$, feature attributions can be computed by integrating the gradient estimator along the path $\boldsymbol{\alpha}(\gamma)$ for $\gamma \in [0,1]$. When taking the straightline path $\boldsymbol{\alpha}(\gamma) = \gamma \cdot \mathbb{1}_p$, which is the only symmetry-preserving path (Sundararajan et al., 2017), the GEFA explainer is derived as follows:

$$\boldsymbol{\xi} := \int_0^1 \boldsymbol{\eta}_{\boldsymbol{\alpha}}(\boldsymbol{x}(\gamma \cdot \mathbb{1}_p)) \, \mathrm{d}\gamma$$

$$= \int_0^1 \mathbb{E}_{\boldsymbol{\pi}(\boldsymbol{\epsilon}|\gamma \cdot \mathbb{1}_p)}[f(\boldsymbol{\epsilon} \circ \boldsymbol{x} \oplus \bar{\boldsymbol{\epsilon}} \circ \mathring{\boldsymbol{x}}) \cdot (\frac{\boldsymbol{\epsilon}}{\gamma} - \frac{\bar{\boldsymbol{\epsilon}}}{1-\gamma})] \, \mathrm{d}\gamma \tag{7}$$

In practice, equation 7 can be approximated with a Monte-Carlo estimator, given a budget of $n$ queries:

$$\boldsymbol{\xi} \approx \frac{1}{n} \sum_{\gamma \sim \mathcal{U}_{[0,1]}} \sum_{\boldsymbol{\pi}(\boldsymbol{\epsilon}|\gamma \cdot \mathbb{1})} f(\boldsymbol{\epsilon} \circ \boldsymbol{x} \oplus \bar{\boldsymbol{\epsilon}} \circ \mathring{\boldsymbol{x}}) \cdot (\frac{\boldsymbol{\epsilon}}{\gamma} - \frac{\bar{\boldsymbol{\epsilon}}}{1-\gamma}) \tag{8}$$

**Theorem 1.** *GEFA satisfies the property of Completeness, Sensitivity, Insensitivity, Linearity, and Symmetry.*

Appendix A.2 details these properties and the corresponding proof derived from the gradient estimation perspective following equation 7. On top of the proved properties, we surprisingly find that GEFA, an approach derived from a proxy gradient estimator, is an alternative to compute Shapley Values as stated in Theorem 2.

**Theorem 2.** *Feature attributions determined by GEFA are exactly Shapley Values.*

The claim in Theorem 2 is mathematically rigorously proved, please refer to Appendix A.1 for further details. Being an unbiased calculator of Shapley Values also explains the many properties hold by GEFA.

While also producing an unbiased approximation of Shapley Values, GEFA differs from other sampling-based attempts by simplifying the sampling process. Concretely, the computation of equation 8 does not rely on marginal contributions, thus avoiding potential information wastes during approximation. Let $\boldsymbol{z_S}$ denote a query with $\boldsymbol{S}$ being the set of indices corresponding to the present features. In GEFA, each query $\boldsymbol{z_S}$ contributes to the attribution estimates of any feature $x_i, \forall i \in \{1, 2, \ldots, p\}$, regardless of the existence of a paired sample $\boldsymbol{z_{S \cup \{i\}}}$ (for $i \notin \boldsymbol{S}$) or $\boldsymbol{z_{S \setminus \{i\}}}$ (for $i \in \boldsymbol{S}$) that would be required for computing marginal contributions. Algorithm 1 summarizes the overall explanation scheme derived from equation 8.

---

**Algorithm 1** GEFA Explanation Scheme

---

**Input:** $\boldsymbol{x}$: the explicand; $\mathring{\boldsymbol{x}}$: the baseline;
**Output:** $\boldsymbol{\xi}$: feature attribution scores;
  1: $\boldsymbol{\xi} = \mathbb{0}_p$                         *# Estimator initialization*
  2: **while** Query budget available **do**
  3:     $\gamma \sim \mathcal{U}_{[0,1]}$            *# Proxy path point sampling*
  4:     $\boldsymbol{\epsilon} \sim \boldsymbol{\pi}(\cdot|\gamma \cdot \mathbb{1}_p)$          *# Mask sampling*
  5:     $\boldsymbol{z} = \boldsymbol{\epsilon} \circ \boldsymbol{x} \oplus \bar{\boldsymbol{\epsilon}} \circ \mathring{\boldsymbol{x}}$       *# Query construction*
  6:     $\boldsymbol{\xi} = \boldsymbol{\xi} + \frac{1}{n} \cdot f(\boldsymbol{z}) \cdot (\frac{\boldsymbol{\epsilon}}{\gamma} - \frac{\bar{\boldsymbol{\epsilon}}}{1-\gamma})$    *# Observation collection*
  7: **end while**
  8: **return** $\boldsymbol{\xi}$

---

## 4.3 Variance Reduction

Deriving the explainer from a score-function gradient estimator allows the application of variance reduction techniques in the gradient estimation literature. Specifically, we construct a control

variate that reduces the estimation variance under the assumption that the target model outcomes are correlated with the number of present features, denoted by $|\boldsymbol{\epsilon}| = \sum_{i=1}^{p} \epsilon_i$. Assumption 1 formally states the condition required for the *validity* of the designed control variate.

**Assumption 1.** For any explicand-baseline pair that satisfies $f(\boldsymbol{x}) \neq f(\mathring{\boldsymbol{x}})$, the correlation between the number of present features and the corresponding model outcomes should be non-zero.

In practice, we argue that the above assumption generally holds for any properly trained model that makes predictions based on (either appropriate or inappropriate Geirhos et al. (2020)) evidence from its inputs. This is because a higher ratio of presented features induces a higher likelihood of including relevant components, thus a convergence toward the prediction result $f(\boldsymbol{x})$. Based on this assumption, the control variate is constructed as a function of $|\boldsymbol{\epsilon}|$:

$$
h(|\boldsymbol{\epsilon}|) = \begin{cases} 0 & \text{if } |\boldsymbol{\epsilon}| = p \\ |\boldsymbol{\epsilon}|/p & \text{else} \end{cases} \tag{9}
$$

Adding the control variate weighted by a fixed hyperparameter $\beta$ to the target function gives:

$$
\tilde{f}(\boldsymbol{\epsilon} \circ \boldsymbol{x} \oplus \bar{\boldsymbol{\epsilon}} \circ \mathring{\boldsymbol{x}}) = f(\boldsymbol{\epsilon} \circ \boldsymbol{x} \oplus \bar{\boldsymbol{\epsilon}} \circ \mathring{\boldsymbol{x}}) - \beta \cdot h(|\boldsymbol{\epsilon}|) \tag{10}
$$

Replacing $f(\cdot)$ in equation 7 accordingly with the updated $\tilde{f}(\cdot)$ yields the variant GĒFA:

$$
\tilde{\boldsymbol{\xi}} = \int_0^1 \mathbb{E}_{\boldsymbol{\pi}(\boldsymbol{\epsilon}|\gamma \cdot \mathbb{1}_p)}[\tilde{f}(\boldsymbol{\epsilon} \circ \boldsymbol{x} \oplus \bar{\boldsymbol{\epsilon}} \circ \mathring{\boldsymbol{x}}) \cdot (\frac{\boldsymbol{\epsilon}}{\gamma} - \frac{\bar{\boldsymbol{\epsilon}}}{1 - \gamma})] \, \mathrm{d}\gamma \tag{11}
$$

**Theorem 3.** *The unbiasness of $\tilde{\boldsymbol{\xi}}$ remains intact after the introduction of the control variate $h(\cdot)$.*

Appendix A.3 provides the proof of Theorem 3, along with the derivation and further details of $h(\cdot)$. The variance reduction effect is optimized when the weighting hyperparameter $\beta = \text{Cov}(f, h)/\text{Var}(h)$. While the variance of the control variate can be computed in closed form, the covariance, albeit not explicitly given, can be empirically estimated (Mohamed et al., 2020) with existing queries for attribution estimation.

## 4.4 RELATION TO INTEGRATED GRADIENTS

Since the proposed method is built upon estimated gradients, this section further explores its relationship to IG[2] that utilizes actual gradients. The equivalence between GEFA and IG does not hold when both take a straightline path, as GEFA's path is constructed in the proxy space, which differs from the original feature space. However, the relation becomes clearer when both explainers follow a monotonic path along the edges of their respective spaces. Along an edge path, integration moves step-by-step from one vertex $\boldsymbol{z}_{\boldsymbol{S}}$ in the feature/proxy space to an adjacent vertex $\boldsymbol{z}_{\boldsymbol{S} \cup \{i\}}$ that differs in only one feature.

**Theorem 4.** *GEFA and IG are equivalent when taking the same edge path. Averaging their results over all $p!$ unique edge paths converges to the outcome of GEFA that follows the straightline path in the proxy space.*

It can be easily shown that, when following the same permutation order, GEFA and IG both compute the marginal contribution of a feature $x_i$, namely $f(\boldsymbol{z}_{\boldsymbol{S}}) - f(\boldsymbol{z}_{\boldsymbol{S} \cup \{i\}})$, conditioned on a set of present features $\{x_j | j \in \boldsymbol{S}\}$. Given the fact that GEFA is an unbiased estimator of Shapley Values, concluding Theorem 4 is not surprising – averaging marginal contributions is the typical solution for determining Shapley Values. Please refer to Appendix A.4 for the detailed proof. The close relationship between IG and Shapley Values is consistent with previous claims by Sundararajan & Najmi (2020). Furthermore, Theorem 4 motivates the choice of the straightline path along the diagonal of the proxy space, converting the problem of averaging the estimates of several edge paths to estimating attributions on one specific path.

---

[2]By considering IG, we omit the practical difficulty that discrete features are usually not differentiable in their original forms, thus requiring additional pre-/post-processing steps.

## 5 EXPERIMENTS

### 5.1 EXPERIMENTAL SETTING

To show GEFA's applicability under various scenarios, we consider the most representative tasks involving discrete and continuous features: text and image classifications.

*Dataset*: The *Amazon* reviews polarity (McAuley & Leskovec, 2013) is adopted for setting up a sentiment analysis task for text classification. The review texts in the dataset include customer reviews of products with a maximal length of 512 tokens, labeled as either positive or negative. As for image classification, we consider *ImageNet* (Russakovsky et al., 2015), a dataset sets up a multi-class classification task in high-dimensional input feature space, posing challenges to black-box explainers that derives feature attributions by querying.

*Classifier*: We fine-tune a publicly available pretrained version of BERT [3] on the *Amazon* review dataset. For ImageNet, a pre-trained version of InceptionV3[4] is adopted without further training. The choice of the two models involves attention mechanisms and the traditional convolution layer, which are the most popular components in current neural network designs but have very different architectures from each other, with the purpose of demonstrating that GEFA's explanation quality is independent of specific model structures.

*Evaluation via manipulation*: Despite explainability being a widely studied topic, there is yet no consent for the quantitative evaluation of explanation quality due to the lack of ground truth explanations (which we are seeking). Compromising to the practical difficulty, a popular evaluation scheme, evaluation via deletion (Samek et al., 2016), quantifies explainer's performance indirectly by summarizing the effectiveness of feature removal guided by explanations. Following the intuition that deleting relevant features should induce significant changes in prediction results, the evaluation scheme repeatedly removes features in descending order according to their attribution scores. The area over the curve drawn by the sequence of prediction outcomes quantifies explanation quality. A larger area indicates a more informative explanation that boosts the impact of the deletion process. Formally, let $\boldsymbol{x}^{(k)}$ denote a manipulated version of the explicand with a number of $k$ features removed, the normalized AOPC (area over perturbation curve) is computed by:

$$\text{nAOPC} = \frac{1}{p} \sum_{k=1}^{p} (1 - \frac{f(\boldsymbol{x}^{(k)})}{f(\boldsymbol{x})})$$

*Competitors*: We consider several feature attribution methods closely related to the proposed method, including two gradient-based approaches assuming white-box access and three black-box explainers:

- **VG** (Vanilla Gradient): an approach interpreting raw gradients directly as explanations
- **IG** (Integrated Gradients): a method integrating actual gradients along a straightline path
- **KSHAP** (KernelSHAP): a Shapley Value estimator built upon weighted linear regression
- **PSHAP** (PartitionSHAP): a variant of sampling-based estimator computing Shapley Values recursively through a hierarchy of features (Chen et al., 2023)
- **GEEX** (Gradient-Estimation-based Explanation): a black-box approach deriving explanations based on estimated gradients

The selected competitors are evaluated following the above evaluation scheme and compared to the two variants of the proposed methods: GEFA and GÊFA, denoting the version without and with the control variate respectively. In addition to the listed explainers, a random feature remover (abbreviated as **Random**) is considered a baseline competitor. It removes features on a random basis imitating that there is no explanatory information. Any explainer that delivers valid explanations should achieve a higher nAOPC score than random removal. While evaluation via deletion has been a widely adopted scheme for assessing explanation quality, concerns have been raised regarding the validity of its results as the recursive deletion process may shift the manipulated explicand away from the target data manifold (Hooker et al., 2019; Jethani et al., 2021). In Appendix B, we provide a more

---

[3]https://huggingface.co/docs/transformers/model doc/bert
[4]https://pytorch.org/vision/stable/models/inception.html

detailed discussion on the validity of the adopted evaluation scheme and demonstrate its alignment with the retraining scheme (Hooker et al., 2019), which circumvents the out-of-distribution concern.

## 5.2 EXPLAINING TEXT CLASSIFIER

When applying feature attributions to text classifiers, black-box approaches like GEFA are more flexible in terms of representing feature absence, as they construct synthetic instances in the original text space for querying. Unlike models for other classification tasks, text classifiers commonly accept inputs with variable lengths, simplifying the definition of absence. When taking an empty token as the baseline, feature absence is represented by the removal of a corresponding feature, providing a more explicit representation of feature absence – a feature is not part of the input – instead of replacing the original value with some manually defined baseline value.

On the other hand, the white-box approaches relying on back-propagation stick to the absence definition as the replacement by some default value, because back-propagation for exact gradient measurement always requires a placeholder in the input as the destination of the propagation process. Specifically, feature absence is modeled by a zero embedding vector for both VG and IG. Furthermore, given texts as sequences of discrete features are not directly differentiable, approaches based on actual gradients require at least one pre-processing step to acquire the embeddings for back-propagation and summarize the embedding-level attributions in the form of token-level results through post-processing for deriving human-comprehensible explanations.

We employ two deletion operations for the evaluation scheme: *embedding reset* and *token removal*, corresponding to the distinct representations of feature absence during the explanation processes. *Embedding reset* conducts deletion by setting the embedding vector of a token being removed to a zero vector, aligning with the absence representation adopted by back-propagation-based methods. *Token removal* wipes the presence of a token completely by replacing it with an empty token. Table 1 presents the nAOPC scores of the competitors tested with both deletion operations. Each row in the table corresponds to the nAOPCs for the respective deletion type indicated in the first column. For the black-box explainers, given the relatively smaller feature space, we empirically set a query budget of 500. In the case of IG, the gradient is integrated over 50 interpolated points in the embedding space. Please note that GEEX is excluded from this part of the evaluation due to its incompatibility with models operating on discrete feature space, as previously discussed in Section 1.

Notably, the explanations by VG barely deliver any valid information as evidenced by its performance, which is at the level of random removal in both deletion settings. This observation suggests that directly interpreting gradients as explanations is inappropriate since the raw gradient itself only reveals a model's local sensitivity to a feature, which does not necessarily associate with the feature's contribution to a prediction. The qualitative example in Figure 1 showcases the failure of VG to capture relevant features in contrast to IG and GẼFA. While there are disagreements in attributions between IG and GẼFA, their explanations agree on the main evidence for a positive prediction; whereas VG produces a contradictory result by identifying 'pain' as a positively contributing feature and puts a stop word 'that' as import evidence in sentiment analysis, which appears less sensical.

Among the group of black-box explainers, GẼFA achieves the best performance over other sampling-based Shapley Value estimators. We attribute the improvement to the information waste minimization during the estimation and the variance reduction led by the designed control variate. The comparison between both GEFA variants highlights the effectiveness of the control variate, which follows a simple intuition. Parallel to the comparison among black-box approaches, GẼFA, despite being constrained with query-level access, demonstrates performance comparable to IG in the embedding reset setting and even surpasses the white-box explainer when tested with token removal. Given GẼFA's improved performance through variance reduction, it is reasonable to infer that the proposed method could outperform IG in both settings if the estimator can be further strengthened, for instance, by increasing the query budget. The distinct *absence representations* is considered the main source of the observed performance differences. We argue that setting a feature to a default value does not faithfully reflect the status of a feature being absent, as the specific choice of baseline can introduce inductive bias. This concern is particularly relevant to feature absence modeling in language models, where a natural definition of absence – token removal – is easily accessible.

Table 1: The nAOPCs reported on BERT fine-tuned for *Amazon* reviews, higher is better.

| Deletion Type | VG | IG | KSHAP | PSHAP | GEFA | GẼFA | Random |
|---|---|---|---|---|---|---|---|
| Embed. reset | 0.2129 | **0.6622** | 0.5446 | 0.6358 | 0.6275 | 0.6482 | 0.2113 |
| Token removal | 0.1823 | 0.6677 | 0.6014 | 0.6592 | 0.7120 | **0.7366** | 0.1908 |

*The overall best performances are in **bold** and the highest scores among black-box explainers are underlined.

ive got a lamp in the corner of my room behind my desk thats a complete pain in the arse to turn on and off. ive been using this with the lamp for a month now and it works perfectly. added a little ve lcro and now i have a light switch where ever i want. under my desk, shelf, etc.

(a) VG

ive got a lamp in the corner of my room behind my desk thats a complete pain in the arse to turn on and off. ive been using this with the lamp for a month now and it works perfectly. added a little ve lcro and now i have a light switch where ever i want. under my desk, shelf, etc.

(b) IG

ive got a lamp in the corner of my room behind my desk thats a complete pain in the arse to turn on and off. ive been using this with the lamp for a month now and it works perfectly. added a little ve lcro and now i have a light switch where ever i want. under my desk, shelf, etc.

(c) GEFA

Figure 1: Feature attributions for BERT derived from three selected explainers. The results are visualized by attribution maps, where a blue/red background color indicates a contribution to the positive/negative sentiment with the color intensity reflecting the amplitude of the attribution score.

## 5.3 EXPLAINING IMAGE CLASSIFIER

We repeat the same evaluation to assess the quality of explanations for image classification results. The query budget of the black-box approaches is increased to 5000 due to the considerably larger input feature space, having a size of $299 \times 299$. KSHAP is excluded from this part of the evaluation as solving the linear regression requires a query budget matching the dimensionality of the input feature space, which is less practical for models taking high-dimensional inputs. Since image classifiers cannot process incomplete inputs, feature absence in this context is represented by replacing features with a baseline value. In accordance with the suggestion by Sturmfels et al. (2020), we use a blurred version of the explicand as the baseline.

As shown in Table 2, the performance and relative ranking of the competitors are consistent with the observation from the previous experiment. GẼFA retains competitive performance in the high-dimensional setting compared to the best-performing white-box approach. It is noteworthy that, when explaining the image classifier, the control variate yields a larger performance improvement for GẼFA than in the setting of sentiment analysis. This is found to be caused by a stronger correlation between the control variate and the decision function. In image classification, each feature – a pixel – contributes minorly to the overall prediction and typically possesses less semantic weight contradictory to features in sentiment analysis, where contextual dependencies on specific tokens (such as negation or irony) undermine the validity of Assumption 1 to some extent. With the variance of the control variate remaining constant, the increased amplitude of the covariance between $f(\cdot)$ and $h(\cdot)$ contributes positively to variance reduction as detailed in Appendix A.3, thus enhancing the overall quality of explanations.

Additionally, the comparison between the proposed approach and GEEX, the other gradient-estimation-based method, is worth mentioning. Queries by GEFA, constructed through binarized feature value sampling, induce more significant prediction changes than those created by adding small Gaussian noises, which facilitates more effective gradient estimation. In the experiments, we find that explanations by GEEX are more sensitive to low-level features that are generally informative, such as contours of objects, but they struggle to differentiate which specific class those features contribute to. Examples listed in Figure 2 demonstrate that GẼFA distinguishes features relevant to specific classes, whereas GEEX fails to do so. In the "dog-cat" example, although there are differences in GEEX's explanations between the selected classes, pixels relevant to "dog" are consistently highlighted, whose relationship to the diverse predictions is difficult to comprehend. On the contrary, the explanations by GEFA clearly differentiate the contributions of the same features in different contexts, as indicated by the coloring of the pixels. The pixels representing "dog" and "cat" show conflicting contributions, which is a result of the softmax layer concatenated before the final output layer – the probability increase for one class undermines the other. Similar observations can be obtained in the "rooster-hen" example, where GEEX concentrates on one object and overlooks the fact that the model can distinguish between a rooster and a hen, as demonstrated by GEFA.

Table 2: The nAOPCs reported on InceptionV3 for ImageNet, higher is better.

| Deletion Type | VG | IG | PSHAP | GEEX | GEFA | GẼFA | Random |
|---|---|---|---|---|---|---|---|
| Pixel reset | 0.4570 | **0.8805** | 0.7753 | 0.7952 | 0.8352 | 0.8747 | 0.4003 |

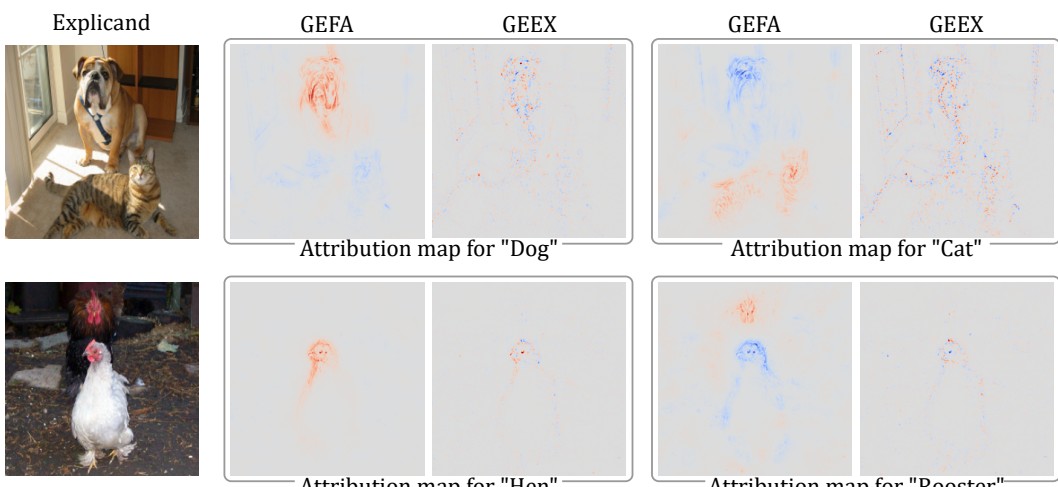

Figure 2: Feature attribution for InceptionV3 showing evidence for prediction as a specific class. Pixels colored in red support the prediction of the targeted class, whereas blue pixels against the prediction. Color intensity indicates the amplitude of attributions.

## 6 CONCLUSION

In this paper, we propose GEFA, a model-agnostic feature attribution framework based on a proxy gradient estimator. By structuring the explanation process in the proxy space, GEFA is generally applicable for explaining arbitrary classifiers, regardless of their input feature types. Backed by rigorous theoretical analysis, the proposed method significantly improves the quality of black-box explanations and, in certain circumstances, even surpasses white-box approaches with a limited query budget. As a general framework, GEFA holds significant potential for integration with existing techniques, further enhancing sampling efficiency (Shrotri et al., 2022; Dhurandhar et al., 2024) and explanation quality (Frye et al., 2020; Heskes et al., 2020).

Although our current focus is on feature attribution for classification tasks, the versatility of GEFA opens avenues for future work, particularly in adapting it to more complicated scenarios, such as explaining multi-modal models like CLIP (Radford et al., 2021) and large language models. These potential adaptions would primarily require reformatting the loss function to handle more complex model outcomes, while the core of the explanation framework remains unchanged.

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

## A    MATHEMATICAL PROOFS

### A.1    PROOF OF GEFA'S EQUIVALENCE TO SHAPLEY VALUES

We start with proving Theorem 2, as the notations introduced during the proof facilitate the proof of the properties listed in Theorem 1. To show that the attributions delivered by GEFA are exact Shapley Values, the goal is to demonstrate the following equivalence:

$$\xi_i = \sum_{\boldsymbol{S} \subseteq \{1,2,\ldots,p\} \backslash \{i\}} \frac{|\boldsymbol{S}|!(p-|\boldsymbol{S}|-1)!}{p!} \cdot (f(\boldsymbol{z}_{\boldsymbol{S} \cup \{i\}}) - f(\boldsymbol{z}_{\boldsymbol{S}})) = Sh_i$$

where $\boldsymbol{z}_{\boldsymbol{S}}$ denotes a query with $\boldsymbol{S}$ being the set of indices corresponding to the present features.

*Proof of Theorem 2.* Let $\boldsymbol{z}_{\boldsymbol{S}}$ be a query, the probability of sampling $\boldsymbol{z}_{\boldsymbol{S}}$ over the integration path is:

$$p(\boldsymbol{z}_{\boldsymbol{S}}|\boldsymbol{x}) = \int_0^1 \gamma^{|\boldsymbol{S}|} \cdot (1-\gamma)^{(p-|\boldsymbol{S}|)} \, \mathrm{d}\gamma$$

For a feature $x_i$, where $i \notin \boldsymbol{S}$, the contribution of the query to the computation of the corresponding attribution, noted as $w_i^{\boldsymbol{z}_{\boldsymbol{S}}}$, is:

$$w_i^{\boldsymbol{z}_{\boldsymbol{S}}} = \int_0^1 \gamma^{|\boldsymbol{S}|} \cdot (1-\gamma)^{(p-|\boldsymbol{S}|)} \cdot f(\boldsymbol{z}_{\boldsymbol{S}}) \cdot (\frac{0}{\gamma} + \frac{1-0}{1-\gamma}) \, \mathrm{d}\gamma$$

$$= -\int_0^1 \gamma^{|\boldsymbol{S}|} \cdot (1-\gamma)^{(p-|\boldsymbol{S}|-1)} \cdot f(\boldsymbol{z}_{\boldsymbol{S}}) \, \mathrm{d}\gamma$$

$$= -\frac{|\boldsymbol{S}|!(p-|\boldsymbol{S}|-1)!}{p!} \cdot f(\boldsymbol{z}_{\boldsymbol{S}}) \qquad \text{(Beta-function)}$$

Similarly, the weight of the query $\boldsymbol{z}_{\boldsymbol{S} \cup \{i\}}$ that differs from $\boldsymbol{z}_{\boldsymbol{S}}$ only in the $i$-th feature is:

$$w_i^{\boldsymbol{z}_{\boldsymbol{S} \cup \{i\}}} = \int_0^1 \gamma^{|\boldsymbol{S}|+1} \cdot (1-\gamma)^{(p-|\boldsymbol{S}|-1)} \cdot f(\boldsymbol{z}_{\boldsymbol{S} \cup \{i\}}) \cdot (\frac{1}{\gamma} + \frac{1-1}{1-\gamma}) \, \mathrm{d}\gamma$$

$$= \frac{|\boldsymbol{S}|!(p-|\boldsymbol{S}|-1)!}{p!} \cdot f(\boldsymbol{z}_{\boldsymbol{S} \cup \{i\}})$$

Summing over all possible combinations of feature presences (excluding $x_i$), yields $\xi_i$:

$$\xi_i = \sum_{\boldsymbol{S} \subseteq \{1,2,\ldots,p\} \backslash \{i\}} w_i^{\boldsymbol{z}_{\boldsymbol{S}}} + w_i^{\boldsymbol{z}_{\boldsymbol{S} \cup \{i\}}}$$

$$= \sum_{\boldsymbol{S} \subseteq \{1,2,\ldots,p\} \backslash \{i\}} \frac{|\boldsymbol{S}|!(p-|\boldsymbol{S}|-1)!}{p!} \cdot (f(\boldsymbol{z}_{\boldsymbol{S} \cup \{i\}}) - f(\boldsymbol{z}_{\boldsymbol{S}}))$$

$$\Leftrightarrow Sh_i$$

$\square$

### A.2    PROOFS OF CLAIMED PROPERTIES

It is not surprising that GEFA aligns with the properties held by Shapley Values as an unbiased calculator. This section details the proof of these properties from the gradient estimator perspective as an alternative to the derivation from the typical computation of Shapley Values in the form of marginal contributions.

### A.2.1    COMPLETENESS AND SENSITIVITY

**Completeness** requires the equivalence between the sum of allocated feature attributions and the difference in prediction results made by full feature presence as stated in equation 1.

*Proof of Completeness.* The contribution of a sample $z_S$ to attribution estimation in GEFA can be divided into two parts, the contribution with a positive sign $w_{i \in S}$ to the present features $\{x_i | i \in S\}$, and the contribution with a negative sign $w_{i \notin S}$ to the absent features. According to equation 8, the contribution is computed by:

$$w_{i \in S} = f(z_S) \cdot \frac{1}{\gamma}$$

$$w_{i \notin S} = -f(z_S) \cdot \frac{1}{1 - \gamma}$$

Considering the likelihood of $z_S$ being sampled, the total positive contribution $w_S^{\oplus}$ can be computed by:

$$
\begin{aligned}
w_S^{\oplus} &= \int_0^1 \gamma^{|S|} \cdot (1 - \gamma)^{(p - |S|)} \cdot \left( \sum_{i \in S} w_{i \in S} \right) \mathrm{d}\gamma \\
&= \int_0^1 \gamma^{|S|} \cdot (1 - \gamma)^{(p - |S|)} \cdot f(z_S) \cdot \frac{|S|}{\gamma} \, \mathrm{d}\gamma \\
&= \frac{(|S| - 1)!(p - |S|)!}{p!} \cdot f(z_S) \cdot |S| \qquad \text{(Beta-function)} \\
&= \frac{|S|!(p - |S|)!}{p!} \cdot f(z_S)
\end{aligned}
$$

Similarly, the total negative contribution is:

$$
\begin{aligned}
w_S^{\ominus} &= -\int_0^1 \gamma^{|S|} \cdot (1 - \gamma)^{(p - |S|)} \cdot f(z_S) \cdot \frac{p - |S|}{1 - \gamma} \, \mathrm{d}\gamma \\
&= -\frac{(|S|)!(p - |S| - 1)!}{p!} \cdot f(z_S) \cdot (p - |S|) \\
&= -\frac{|S|!(p - |S|)!}{p!} \cdot f(z_S)
\end{aligned}
$$

The two parts of contributions cancel out as $w_S^{\oplus} + w_S^{\ominus} = 0$, with the only two exceptions when $S = \emptyset$ or $S = \{1, 2, \ldots, p\}$, whose contribution only has the negative/positive part:

$$w_{\emptyset}^{\oplus} + w_{\emptyset}^{\ominus} = 0 - f(\mathring{x})$$

$$w_{\{1,2,\ldots,p\}}^{\oplus} + w_{\{1,2,\ldots,p\}}^{\oplus} = f(x) - 0$$

Computing the sum of feature attributions by summarizing sample contributions results in:

$$\sum_{i=1}^p \xi = \sum_{S \subseteq \{1,2,\ldots,p\}} (w_S^{\oplus} + w_S^{\ominus}) = f(x) - f(\mathring{x})$$

$\square$

*Sensitivity* is guaranteed by the satisfaction of completeness.

### A.2.2   INSENSITIVITY

**Insensitivity** is also known as *Dummy*, which requires the attribution score to be zero for any feature on which the target model is not functionally dependent. Definition 1 formally describes functional independence.

**Definition 1.** A function is said to be *functionally independent* of a feature if the prediction results are always the same for any sample pair that differs only in that feature.

*Proof of Insensitivity.* Let $x_i$ be the dummy feature, the proxy gradient estimator of that feature on the straightline path is:

$$\eta_{\alpha_i}(\boldsymbol{x}(\gamma \cdot \mathbb{1}_p)) = \mathbb{E}_{\boldsymbol{\pi}(\boldsymbol{\epsilon}|\gamma \cdot \mathbb{1}_p)}[f(\boldsymbol{\epsilon} \circ \boldsymbol{x} \oplus \bar{\boldsymbol{\epsilon}} \circ \mathring{\boldsymbol{x}}) \cdot (\frac{\epsilon_i}{\gamma} - \frac{\bar{\epsilon}_i}{1 - \gamma})]$$

Using $\boldsymbol{\pi}(\boldsymbol{\epsilon}_{\backslash i}|\gamma \cdot \mathbb{1}_{p-1})$ as a shorthand for the feature value sampling process excluding the $i$-th feature, the expectation can be expanded to the following form due to the independent sampling processes of different features:

$$\eta_{\alpha_i}(\boldsymbol{x}(\boldsymbol{\alpha})) = \mathbb{E}_{\boldsymbol{\pi}(\boldsymbol{\epsilon}_{\backslash i}|\gamma \cdot \mathbb{1}_{p-1})}\Big[\mathbb{E}_{\boldsymbol{\pi}(\epsilon_i|\gamma)}[f(\boldsymbol{\epsilon} \circ \boldsymbol{x} \oplus \bar{\boldsymbol{\epsilon}} \circ \mathring{\boldsymbol{x}}) \cdot (\frac{\epsilon_i}{\gamma} - \frac{\bar{\epsilon}_i}{1 - \gamma})]\Big]$$

The condition of functional independence of $x_i$ yields:

$$\eta_{\alpha_i}(\boldsymbol{x}(\boldsymbol{\alpha})) = \mathbb{E}_{\boldsymbol{\pi}(\boldsymbol{\epsilon}_{\backslash i}|\gamma \cdot \mathbb{1}_{p-1})}\Big[\mathbb{E}_{\boldsymbol{\pi}(\epsilon_i|\gamma)}[f(\boldsymbol{\epsilon} \circ \boldsymbol{x} \oplus \bar{\boldsymbol{\epsilon}} \circ \mathring{\boldsymbol{x}})] \cdot \underbrace{\mathbb{E}_{\boldsymbol{\pi}(\epsilon_i|\gamma)}[(\frac{\epsilon_i}{\gamma} - \frac{\bar{\epsilon}_i}{1 - \gamma})]}_{=0}\Big]$$

$$= 0$$

The explainer integrating over $\eta_{\alpha_i}(\boldsymbol{x}(\boldsymbol{\alpha}))$ also produces zero, namely $\xi_i = 0$. $\qquad\square$

### A.2.3 LINEARITY

For any two functions $f_1(\cdot)$ and $f_2(\cdot)$, **Linearity** requires the explanation for the linear composition of the two functions equaling the weighted sum of the separate explanations for them:

$$\boldsymbol{\xi}^{(af_1+bf_2))} = a \cdot \boldsymbol{\xi}^{(f_1)} + b \cdot \boldsymbol{\xi}^{(f_2)}$$

*Proof of Linearity.*

$$\boldsymbol{\xi}^{(af_1+bf_2))} = \int_0^1 \mathbb{E}_{\boldsymbol{\pi}(\boldsymbol{\epsilon}|\gamma \cdot \mathbb{1}_p)}\Big[[af_1(\boldsymbol{\epsilon} \circ \boldsymbol{x} \oplus \bar{\boldsymbol{\epsilon}} \circ \mathring{\boldsymbol{x}}) + bf_2(\boldsymbol{\epsilon} \circ \boldsymbol{x} \oplus \bar{\boldsymbol{\epsilon}} \circ \mathring{\boldsymbol{x}})] \cdot (\frac{\boldsymbol{\epsilon}}{\gamma} - \frac{\bar{\boldsymbol{\epsilon}}}{1 - \gamma})\Big]\,\mathrm{d}\gamma$$

$$= a \cdot \int_0^1 \mathbb{E}_{\boldsymbol{\pi}(\boldsymbol{\epsilon}|\gamma \cdot \mathbb{1}_p)}\Big[f_1(\boldsymbol{\epsilon} \circ \boldsymbol{x} \oplus \bar{\boldsymbol{\epsilon}} \circ \mathring{\boldsymbol{x}}) \cdot (\frac{\boldsymbol{\epsilon}}{\gamma} - \frac{\bar{\boldsymbol{\epsilon}}}{1 - \gamma})\Big]\,\mathrm{d}\gamma +$$

$$b \cdot \int_0^1 \mathbb{E}_{\boldsymbol{\pi}(\boldsymbol{\epsilon}|\gamma \cdot \mathbb{1}_p)}\Big[f_2(\boldsymbol{\epsilon} \circ \boldsymbol{x} \oplus \bar{\boldsymbol{\epsilon}} \circ \mathring{\boldsymbol{x}}) \cdot (\frac{\boldsymbol{\epsilon}}{\gamma} - \frac{\bar{\boldsymbol{\epsilon}}}{1 - \gamma})\Big]\,\mathrm{d}\gamma\Big]$$

$$= a \cdot \boldsymbol{\xi}^{(f_1)} + b \cdot \boldsymbol{\xi}^{(f_2)}$$

$\qquad\square$

### A.2.4 SYMMETRY

In context of feature attribution, **Symmetry** states: given a function $f(\cdot)$ that is symmetric in two variables $x_i$ and $x_j$, the attribution scores of the two features satisfies $\xi_i = \xi_j$ when the explicand-baseline pair holds $x_i = x_j$ and $\mathring{x}_i = \mathring{x}_j$.

*Proof of Symmetry.* Similar to the proof of *Insensitivity*, the *Symmetry* of GEFA originates from the proxy gradient estimator. Let $x_i$ and $x_j$ denote the two symmetric features, their gradient estimators are:

$$\eta_{\alpha_i}(\boldsymbol{x}(\gamma \cdot \mathbb{1}_p)) = \mathbb{E}_{\boldsymbol{\pi}(\epsilon_i|\gamma)}\Big[\mathbb{E}_{\boldsymbol{\pi}(\boldsymbol{\epsilon}_{\backslash i}|\gamma \cdot \mathbb{1}_{p-1})}[f(\boldsymbol{\epsilon} \circ \boldsymbol{x} \oplus \bar{\boldsymbol{\epsilon}} \circ \mathring{\boldsymbol{x}})] \cdot (\frac{\epsilon_i}{\gamma} - \frac{\bar{\epsilon}_i}{1 - \gamma})\Big]$$

$$\eta_{\alpha_j}(\boldsymbol{x}(\gamma \cdot \mathbb{1}_p)) = \mathbb{E}_{\boldsymbol{\pi}(\epsilon_j|\gamma)}\Big[\mathbb{E}_{\boldsymbol{\pi}(\boldsymbol{\epsilon}_{\backslash j}|\gamma \cdot \mathbb{1}_{p-1})}[f(\boldsymbol{\epsilon} \circ \boldsymbol{x} \oplus \bar{\boldsymbol{\epsilon}} \circ \mathring{\boldsymbol{x}})] \cdot (\frac{\epsilon_i}{\gamma} - \frac{\bar{\epsilon}_i}{1 - \gamma})\Big]$$

Given the symmetry between $x_i$ and $x_j$, the inner expectations satisfy:

$$\mathbb{E}_{\boldsymbol{\pi}(\boldsymbol{\epsilon}_{\backslash i}|\gamma \cdot \mathbb{1}_{p-1})}[f(\boldsymbol{\epsilon} \circ \boldsymbol{x} \oplus \bar{\boldsymbol{\epsilon}} \circ \mathring{\boldsymbol{x}})] = \mathbb{E}_{\boldsymbol{\pi}(\boldsymbol{\epsilon}_{\backslash j}|\gamma \cdot \mathbb{1}_{p-1})}[f(\boldsymbol{\epsilon} \circ \boldsymbol{x} \oplus \bar{\boldsymbol{\epsilon}} \circ \mathring{\boldsymbol{x}})], \quad \text{when } \epsilon_i = \epsilon_j$$

It is not difficult to show that sampling of the two features following the same distribution given $x_i = x_j$ and $\mathring{x}_i = \mathring{x}_j$, which induces:

$$\eta_{\alpha_i}(\boldsymbol{x}(\gamma \cdot \mathbb{1}_p)) = \eta_{\alpha_j}(\boldsymbol{x}(\gamma \cdot \mathbb{1}_p))$$

Integrating the estimators having the same outputs along the symmetric path concludes the proof by showing:

$$\xi_i = \int_0^1 \eta_{\alpha_i}(\boldsymbol{x}(\gamma \cdot \mathbb{1}_p)) \, \mathrm{d}\gamma = \int_0^1 \eta_{\alpha_j}(\boldsymbol{x}(\gamma \cdot \mathbb{1}_p)) \, \mathrm{d}\gamma = \xi_j$$

$\square$

### A.3 CONTROL VARIATE

To prove the unbiasedness of $\tilde{\boldsymbol{\xi}}$, we need to show $\tilde{\boldsymbol{\xi}} = \boldsymbol{\xi}$. Applying *Linearity*, we can rewrite $\tilde{\boldsymbol{\xi}}$ as:

$$\tilde{\boldsymbol{\xi}} = \boldsymbol{\xi}^{(f)} + \beta \cdot \boldsymbol{\xi}^{(h)} = \boldsymbol{\xi} + \beta \cdot \boldsymbol{\xi}^{(h)}$$

Now, the goal of the proof can be transformed to:

$$\tilde{\boldsymbol{\xi}} = \boldsymbol{\xi} \iff \boldsymbol{\xi}^{(h)} = \mathbb{0}_p$$

*Proof of Theorem 3.* The attribution of the control variate to the $i$-th feature is:

$$\begin{aligned}
\xi_i^{(h)} &= \int_0^1 \mathbb{E}_{\boldsymbol{\pi}(\boldsymbol{\epsilon}|\gamma \cdot \mathbb{1}_p)}[h(\boldsymbol{\epsilon}) \cdot (\frac{\epsilon_i}{\gamma} - \frac{\bar{\epsilon}_i}{1 - \gamma})] \, \mathrm{d}\gamma \\
&= \sum_{\boldsymbol{\epsilon} \in \{0,1\}^p : \epsilon_i = 0} \frac{|\boldsymbol{\epsilon}|!(p - |\boldsymbol{\epsilon}| - 1)!}{p!} \cdot \left( h(|\boldsymbol{\epsilon}| + 1) - h(|\boldsymbol{\epsilon}|) \right) && \text{(Theorem 2)} \\
&= \sum_{|\boldsymbol{\epsilon}|=0}^{p-1} \binom{p-1}{|\boldsymbol{\epsilon}|} \cdot \frac{|\boldsymbol{\epsilon}|!(p - |\boldsymbol{\epsilon}| - 1)!}{p!} \cdot \left( h(|\boldsymbol{\epsilon}| + 1) - h(|\boldsymbol{\epsilon}|) \right) \\
&= \sum_{|\boldsymbol{\epsilon}|=0}^{p-1} \frac{1}{p} \cdot \left( h(|\boldsymbol{\epsilon}| + 1) - h(|\boldsymbol{\epsilon}|) \right) \\
&= \frac{1}{p} \cdot \left( h(p - 1 + 1) - h(0) \right) && \text{(Telescoping series)} \\
&= 0
\end{aligned}$$

The zero-ness of feature attribution $\xi_i^{(h)}$ concludes the proof:

$$\xi_i^{(h)} = 0, \ \forall i \in \{1, 2, \ldots, p\} \implies \boldsymbol{\xi}^{(h)} = \mathbb{0}_p$$

$\square$

While constructing the control variate for GEFA, we first initialize it as $h(|\boldsymbol{\epsilon}|) = |\boldsymbol{\epsilon}|/p$ based on Assumption 1. To strictly follow the property of unbiasedness, the above analysis derives an additional requirement for the control variate, namely:

$$h(p) = h(0)$$

Integrating the constraint into the control variate delivers the function stated in equation 9. In addition to the selected control variate, Theorem 1 applies to the broader group of functions, which depends solely on $|\boldsymbol{\epsilon}|$ and at the same time satisfies $h(p) = h(0)$. When there are further assumptions to make on the target function, the shape of $h(\cdot)$ can be fine-tuned for a stronger covariance in relation to $f(\cdot)$.

Next, we show the variance reduction effect of the control variate is optimized when:

$$\beta^* = \mathrm{Cov}(f, h)/\mathrm{Var}(h)$$

where the optimal choice of the weighting term is denoted as $\beta^*$.

*Proof of Optimality of $\beta^*$.* Denoting the variance of a gradient estimator for a feature $x_i$ as $\mathrm{Var}(\xi_i)$, the variance of the estimator after the introduction of a control variate is:

$$\mathrm{Var}(\tilde{\xi}_i) = \mathrm{Var}(\xi_i) + \beta^2 \mathrm{Var}(\xi_i^{(h)}) - 2\beta \cdot \mathrm{Cov}(\xi_i, \xi_i^{(h)})$$

The optimal variance reduction effect for $\xi_i$ is achieved when:

$$\beta = \text{Cov}(\xi_i, \xi_i^{(h)})/\text{Var}(\xi_i^{(h)}) \tag{12}$$

Alternative to a feature-specific optimal value, we are also interested in a single value for $\beta$ that maximizes the overall variance reduction effect. To acquire the overall optimum, we first expand the covariance in equation 12:

$$\text{Cov}(\xi_i, \xi_i^{(h)}) = \mathbb{E}[\xi_i \cdot \xi_i^{(h)}] - \mathbb{E}[\xi_i] \cdot \mathbb{E}[\xi_i^{(h)}]$$

$$= \mathbb{E}_{\alpha_i}\Big[\mathbb{E}_{\epsilon_i}[f(\boldsymbol{z}) \cdot h(\boldsymbol{z}) \cdot (\nabla_{x_i} \log \pi(\epsilon_i|\alpha_i))^2]\Big] - \mathbb{E}[\xi_i] \cdot 0 \quad \text{(Unbiasedness of } \boldsymbol{\xi}^{(h)})$$

Please note that we omit the distribution that $\alpha_i$ and $\epsilon_i$ should follow as it does not affect the result of the derivation. For high-dimensional input, the functions $f(\cdot)$ and $h(\cdot)$ have trivial dependencies on a specific feature $x_i$:

$$\text{Cov}(\xi_i, \xi_i^{(h)}) \approx \mathbb{E}_{\alpha_i}\Big[\mathbb{E}_{\epsilon_i}[f(\boldsymbol{z}) \cdot h(\boldsymbol{z})]\Big] \cdot \mathbb{E}_{\alpha_i}\Big[\mathbb{E}_{\epsilon_i}[(\nabla_{x_i} \log \pi(\epsilon_i|\alpha_i))^2]\Big]$$

Similarly, the variance of the control variate estimator can be written as:

$$\text{Var}(\xi_i^{(h)}) \approx \mathbb{E}_{\alpha_i}\Big[\mathbb{E}_{\epsilon_i}[h(\boldsymbol{z})^2]\Big] \cdot \mathbb{E}_{\alpha_i}\Big[\mathbb{E}_{\epsilon_i}[(\nabla_{x_i} \log \pi(\epsilon_i|\alpha_i))^2]\Big]$$

Putting together yields the overall optimal value $\beta^*$:

$$\beta^* = \frac{\mathbb{E}_{\alpha_i}\Big[\mathbb{E}_{\epsilon_i}[f(\boldsymbol{z}) \cdot h(\boldsymbol{z})]\Big] \cdot \mathbb{E}_{\alpha_i}\Big[\mathbb{E}_{\epsilon_i}[(\nabla_{x_i} \log \pi(\epsilon_i|\alpha_i))^2]\Big]}{\mathbb{E}_{\alpha_i}\Big[\mathbb{E}_{\epsilon_i}[h(\boldsymbol{z})^2]\Big] \cdot \mathbb{E}_{\alpha_i}\Big[\mathbb{E}_{\epsilon_i}[(\nabla_{x_i} \log \pi(\epsilon_i|\alpha_i))^2]\Big]}$$

$$= \frac{\mathbb{E}_{\alpha_i}\Big[\mathbb{E}_{\epsilon_i}[f(\boldsymbol{z}) \cdot h(\boldsymbol{z})]\Big] - 0}{\mathbb{E}_{\alpha_i}\Big[\mathbb{E}_{\epsilon_i}[h(\boldsymbol{z})^2]\Big] - 0}$$

$$= \text{Cov}(f, h)/\text{Var}(h)$$

$$\square$$

Taking the optimal $\beta^*$, the variance reduction effect depends on the covariance between $f(\cdot)$ and $h(\cdot)$, which motivates Assumption 1:

$$\text{Var}(\xi_i) - \text{Var}(\tilde{\xi}_i) = \text{Cov}(f, h)$$

### A.4 EQUIVALENCE TO IG

*Proof of Theorem 4.* To complete the proof, we first show that both GEFA and IG produce marginal contributions along edge paths.

Recalling that an edge path always moves from one vertex $\boldsymbol{z}_{\boldsymbol{S}}$ to an adjacent vertex that differs $\boldsymbol{z}_{\boldsymbol{S}\cup\{i\}}$ in only the $i$-th feature along edges, the goal is simplified prove that they are calculators of the marginal contribution conditioned on the presence of features $\{x_j | j \in \boldsymbol{S}\}$ for each segment of a path. For the $i$-th segment on an edge path with $\boldsymbol{S}$ denoting the preceding vertices, IG produces:

$$\xi_i^{\text{IG}} = \int_{\boldsymbol{z}_{\boldsymbol{S}}}^{\boldsymbol{z}_{\boldsymbol{S}\cup\{i\}}} \frac{\partial f(\boldsymbol{x})}{\partial x_i} \, \mathrm{d}\boldsymbol{x}$$

$$= f(\boldsymbol{z}_{\boldsymbol{S}\cup\{i\}}) - f(\boldsymbol{S})$$

As the path for GEFA is created in the proxy space, we denote the two proxy vertices on the $i$-th segment by $\boldsymbol{x}(\boldsymbol{\alpha}_{\boldsymbol{S}})$ and $\boldsymbol{x}(\boldsymbol{\alpha}_{\boldsymbol{S}\cup\{i\}})$ for preciseness. The notation $\boldsymbol{\alpha}_{\boldsymbol{S}}$ is analogous to $\boldsymbol{z}_{\boldsymbol{S}}$, which represents:

$$\alpha_i = \begin{cases} 1 & \text{if } i \in \boldsymbol{S} \\ 0 & \text{if } i \notin \boldsymbol{S} \end{cases}$$

When following the same permutation order, GEFA produces the same marginal contribution as IG for the $i$-th segment:

$$\xi_i^{\text{GEFA}} = \int_{\boldsymbol{\alpha_S}}^{\boldsymbol{\alpha_{S \cup \{i\}}}} \mathbb{E}_{\pi(\epsilon_i | \alpha_i)}[f(\boldsymbol{z}) \cdot (\frac{\epsilon_i}{\alpha_i} - \frac{\bar{\epsilon}_i}{1 - \alpha_i})] \, \mathrm{d}\boldsymbol{\alpha}$$
$$= f(\boldsymbol{z}_{\boldsymbol{S} \cup \{i\}}) - f(\boldsymbol{z}_{\boldsymbol{S}})$$
$$\Leftrightarrow \xi_i^{\text{IG}}$$

Please note that, for GEFA, the only feature value in $\boldsymbol{z}$ that may vary during the sampling on the $i$-th segment is $z_i$. The remaining features are deterministic as their corresponding proxy variables are either 0 or 1 depending on whether they are included in the preceding vertices $\boldsymbol{S}$, namely to take either the baseline or explicand value with hundred percent probability.

As both explainers deliver marginal contributions along edge paths, the claim in Theorem 4 becomes obvious as it describes the typical computation of Shapley Values. □

## B  VALIDITY OF THE EVALUATION SCHEME

In this work, we adopted the evaluation via deletion approach, a commonly used method for accessing explanation quality. The employment of this evaluation scheme spans from the early stages of explainability research (Samek et al., 2016; Montavon et al., 2018) to the most recent studies (Cai & Wunder, 2024; Muzellec et al., 2024). One of its key advantages is that it does not require retraining during evaluation, thereby offering a static environment for efficient explanation quality assessment. While the out-of-distribution issue raises concern about the validity of the evaluation results, our results regarding the performance of the random feature remover, as reported in Table 1 and Table 2, alleviate this concern. The significantly lower nAOPC scores of random removal demonstrate that simply shifting the input away from the underlying data manifold does not effectively affect model performance.

Hooker et al. (2019) highlighted the issue that out-of-distribution manipulations can trigger unexpected model behaviors. As an alternative to the traditional deletion scheme, they proposed the remove and retraining (ROAR) scheme. This approach involves removing a proportion of features with the highest attribution scores for each instance in the dataset, followed by retraining the model on the manipulated dataset. The resulting model performance is then used as a reflection of the explainer's effectiveness. ROAR assessments reported that many popular attribution methods "*are not better than a random designation of feature importance*", contradicting the conclusions drawn from the traditional deletion scheme. This conflict with the widely approved effectiveness of the tested approaches, such as IG, prompted our further investigation, which revealed the following:

- The root cause of the misalignment between observations from different evaluation perspectives is the **presence of residual features** with negative attributions during the retraining process;
- After a single justified adjustment to ROAR, the two evaluation schemes yield consistent results.

### B.1  THE "SIGN" ISSUE

In the context of feature attribution, a positive attribution score indicates a positive contribution to the prediction result, whereas a negative score, rather than indicating irrelevance, represents a negative association with the decision. Failing to remove negatively contributing features preserves task-relevant information, which the model can reorganize during retraining to improve accuracy. A qualitative example of residual "negative" information in ROAR is illustrated through the visualized pixel removal process of IG in Figure 1 by Hooker et al. (2019). This "sign" issue also explains the effective manipulation by SG-SQ and VarGrad, as these methods provide unsigned attributions, thereby ensuring the removal of all informative features.

To mitigate the assessment distortion caused by retained negative information, we argue that a modification to ROAR is necessary for a more faithful reflection of explainer performance: instead of

Table 3: Performance of explainers in different settings

| Competitors | In Accuracy (%) | | | nAOPC ↑ |
|---|---|---|---|---|
| | ROAR ↓ | ROAR-abs ↓ | KEAR ↑ | |
| IG | 77.20 | **62.80** | 89.60 | **40.82** |
| PSHAP | 79.25 | 76.75 | 84.30 | 39.56 |
| GEFA | 82.35 | 68.45 | **89.95** | 40.79 |
| Random | **71.30** | 71.30 | 71.30 | 35.07 |

↓: lower is better; ↑: higher is better

removing features for retraining, the top-ranked features should be retained. This "keep and retrain" (KEAR) approach reframes the evaluation question as:

- "Does an explanation method effectively identify relevant features?"

An effective explanation method should capture the relevant information learned by the target model, facilitating higher accuracy of the retrained model with the same portion of retained information.

### B.2 RESULTS ON ROAR AND KEAR

To verify the above discussion, we conducted experiments following the retraining scheme. Specifically, we fine-tuned EfficientNet-B0[5] on the Cats vs Dogs dataset (Elson et al., 2007) and created copies of the dataset with explanation-guided manipulation. These manipulated datasets were then used for retraining the pre-trained model to assess the quality of explanations. Without loss of generality, we downsampled the dataset into $2000/400/400$ partitions for training, validation, and test sets for efficiency. EfficientNet-B0 achieved an accuracy of $99.40\%$ on the downsampled dataset after fine-tuning. Based on the fine-tuned EfficientNet-B0, we derived explanations for images in all three partitions with three competitors: IG, PSHAP, and GEFA. Features for each instance were ranked in descending order according to their attribution scores. Similar to the traditional deletion scheme, we adopted the random feature remover as a baseline reference for evaluating the effectiveness of the competitors.

The top $90\%$ of features were **removed** for the ROAR test, whereas the top $10\%$ of features were **kept** for the KEAR test. To highlight the "sign" issue, we also performed feature ranking based on the absolute values of their attribution scores for the removal test, referred to as ROAR-abs (remove and retrain — based on absolute attribution score). To minimize the potential impact of randomness during the training process, we independently retrained 5 models on each manipulated dataset and reported the averaged accuracies of the five models under different settings. It is noteworthy that lower retraining accuracy indicates better explanation quality in the removal tests (ROAR and ROAR-abs), whereas higher accuracy reflects superior explainer performance in KEAR. Results from the desgiend experiments are presented in Table 3. For random removal, the same figures are reported across the three retraining settings as the proportions of remaining features are identical in all cases, i.e. $10\%$.

In the ROAR test, all explanation methods show minor manipulation impacts due to the previously discussed "sign" issue and fail to excel random removal. By contrast, the ROAR-abs test demonstrates that removing residual negative information enhances the effectiveness of manipulation, providing indirect evidence for the "sign" issue. However, the origin of negative attributions is complex and influenced by various factors, e.g. the baseline choice and a feature's association with the prediction function. Ranking features based on the absolute value of their attribution scores may unnecessarily include irrelevant features, as it cannot distinguish between the different causes of a negative sign, rendering this approach a suboptimal solution.

After addressing the distortion caused by residual information, the KEAR test offers a more faithful assessment of explanation quality. The success of the explainers in identifying the most informative features results in relatively high classification accuracy with only $10\%$ of features retained. Notably, our findings closely align with the observations by Hooker et al. (2019). Their results of exactly the same experiment, presented in Table 2 on page 17 (the first four columns), also demonstrate the

---

[5]https://pytorch.org/vision/main/models/efficientnet.html

superiority of IG, consistent with its widely recognized effectiveness under the traditional deletion scheme. The last column of Table 3 presents the nAOPC scores obtained following the traditional deletion approach. The KEAR results, alongside the nAOPCs, show that the retraining scheme and recursive deletion scheme are parallel evaluation options rather than contradictory approaches. While the metrics employed by the two schemes differ in scale, leading to difficulties in direct numerical comparisons, the consistency in relative rankings within each test provides a meaningful reference of nAOPC as a valid metric.

