# OpenReview forum: "A General Feature Attribution Framework under a Black-box Setting"
_ICLR.cc/2025/Conference — Submitted to ICLR 2025_

### Official Review · Reviewer_ALce · 2024-10-23

**Soundness:** 3
**Presentation:** 2
**Contribution:** 3
**Rating:** 6
**Confidence:** 3

**Summary:**

The paper proposes a new framework for feature attribution that can be applied across modalities. Their main idea is use a proxy variable to control feature perturbations with the final outcome being an attribution score through averaging of the results of the perturbations. The proposed approach is for the more general black-box setting. They also theoretically show that their approach satisfies desirable properties such as completeness, insensitivity, linearity, symmetry.

**Strengths:**

- Important problem
- Interesting approach using proxy variables
- Prove that their approach satisfies multiple desirable properties an explanation should have
- Reported empirical results are promising

**Weaknesses:**

- Some related work missing
- Presentation could be improved
- Timing and sample complexity experiments missing
- Experiments on only 2 datasets using simple models
- Not clear how to choose base values
- Only one metric used for evaluation

**Questions:**

I think the approach is interesting, however, I have some concerns with the current manuscript.

*Missing related work:* There are a lot of works on model explanation in the black-box setting also considering linear (proxy) explanation functions (i.e. variants of LIME and SHAP), like the current manuscript, that must be discussed in my opinion. I point to some of these works in [1-7] below.

*Improved presentation:* Equations 6 and 8 seem to be the critical equations in terms of the algorithm to estimate attributions. However, this is lost in the text. Maybe an algorithm can be added using the latex algorithm environment to highlight the main contribution. This will also make clear the computational burden of the algorithm.

*Computational complexity:* The method seems as computationally intensive as SHAP. It would be nice to have some timing comparisons in the experimental section as well as some convergence plots in terms of how the accuracy of attributions approaches the best achieved accuracy with increasing number of samplings. It is not clear intuitively to me why this is a better approach than existing ones like SHAP etc.


*More challenging experiments:* The experiments use small and somewhat outdated models such as BERT. Given the plethora of capable LLMs (viz. mistral, llamma series, etc.) it would be nice to have experiments on more state-of-the-art models. Also only 2 datasets are used. How about at least experimenting on the GLUE benchmark for the LLMs.

*Base values and evaluation:* Not clear how the base values are chosen for the method. I understand that this is an issue for even SHAP etc. but I would like some description of how they chose those in their experiments. Evaluating using ablations is fine. However, there are other metrics for explanation evaluation such as stability, fidelity, etc. [8].

[1] C. J. Anders, P. Pasliev, A.-K. Dombrowski, K.-R. Muller, and P. Kessel. Fairwashing explanations with off-manifold detergent. ICML, 2020.

[2] A. Dhurandhar, K. Ramamurthy, K. Ahuja, and V. Arya. Locally invariant explanations: Towards stable and unidirectional explanations through local invariant learning. NeurIPS, 2023.

[3] T. Botari, F. Hvilshøj, R. Izbicki, and A. C. P. L. F. de Carvalho. Melime: Meaningful local explanation for machine learning models, 2020.

[4] C. Frye, C. Rowat, and I. Feige. Asymmetric shapley values: incorporating causal knowledge into model-agnostic explainability. NeurIPS, 2020.

[5] T. Heskes, E. Sijben, I. G. Bucur, and T. Claassen. Causal shapley values: Exploiting causal knowledge to explain individual predictions of complex models. NeurIPS, 2020.

[6] A. Dhurandhar, K. Ramamurthy, K. Shanmugam. Is this the right neighborhood? Accurate and Query Efficient Model Agnostic Explanations. NeurIPS, 2022.

[7] A. Shrotri, N. Narodytska, A. Ignatiev, J. Marques-Silva, K. S. Meel, and M. Vardi. Constraint-driven explanations of black-box {ml} models, 2021.

[8] QV Liao, Y Zhang, R Luss, F Doshi-Velez and A Dhurandhar. Connecting Algorithmic Research and Usage Contexts: A Perspective of Contextualized Evaluation for Explainable AI. HCOMP, 2022.

---

> ### Author Response · Authors · 2024-11-23
> **Response to Reviewer ALce**
>
> We sincerely appreciate the constructive comments by the reviewer, and we are pleased that our efforts in developing a novel framework for deriving black-box explanations have been well received.
>
> **Missing related work**: First, we would like to thank the reviewer for the detailed list of the suggested references, some of which we were already familiar with due to their significance. The blooming of research in the field of explainability has led to numerous impactful works, and while we aim to include as much as possible, we are constrained by the page limitation. As a result, we narrowed the scope of related work to align closely with the core focus of this paper:
>
> - Gradient-based approaches, which deliver results that we aim to reproduce under a black-box setting;
> - SHAP-family methods, which have a direct and natural connection to our work through Shapley Values.
>
> That said, we agree that refining the related work section to incorporate the suggested references will enhance the manuscript. Specifically, we believe that discussing LIME and SHAP variants will be beneficial in highlighting our framework’s contributions and its distinctions from previous efforts. Parallel to previous efforts in extending the existing frameworks like LIME and SHAP, this manuscript proposes a novel, self-contained framework that is complemented by axiomatic properties.
>
> Furthermore, beyond refining the related work section, we recognize that ideas in the suggested references, especially the idea of refining the sampling process [1, 4, 5] and applying prior casual knowledge [2, 3], are complementary to our proposal. We will discuss these ideas as potential future directions in the conclusion section.
>
> **Improved presentation**: We are currently working on incorporating an algorithm block for a clearer overview of the proposed method. In addition, we will include a diagram to visualize the overall explanation scheme. If space permits, these additions will be included in the body of the updated version; otherwise, they will be added to the appendix and referenced in the main text with a clear statement.
>
> **Computational complexity and effectiveness**: The reviewer is right about GEFA and SHAP having the same computational complexity. The performance improvement of our method stems from two factors:
>
> - Elimination of information waste: Compared to traditional sampling-based approaches, where each query-pair contributes only to the marginal contribution of a specific feature, GEFA leverages each query to estimate the attributions for all features. The detailed statement regarding this point can be found in lines 250-252.
> - Application of control variate: The carefully designed control variate reduces the estimation variance, thereby improving the performance of GEFA without introducing bias. The effectiveness of the control variate is theoretically proved in Appendix A.3, and its empirical benefits are shown through the experimental results comparing GEFA with and without the control variate.
>
> **Baseline value**: The baseline choice is a crucial topic for feature attribution methods, as it largely affects the explanation results. In this work, we use blurred images as the baseline for image data following the suggestion by [6], which conducted a thorough study on the influence of baseline choice. For textual data, we define the baseline token as an empty token, aligning with the natural definition of absence. Replacing a token with an empty token removes the corresponding feature when it is switched off during sampling. While this paper focuses on proposing a novel framework for black-box explanation, an in-depth exploration of the impact of baseline selection lies beyond the scope. That said, we agree that the topic of baseline selection requires further investigations, as it is a general challenge to existing explanation methods.
>
> **Further Evaluation**: We understand the reviewer’s concern about the scope of the evaluation, and would like to refer to our discussion with Reviewer UNFo, where we evaluate the proposed method under the ROAR framework. The results of the additional experiment align with the explanation quality quantified by nAOPC, demonstrating the validity of the chosen metric.

---

> ### Author Response · Authors · 2024-11-23
> **Continued response to detailed points**
>
> **More challenging experiments**: Following the reviewer’s suggestion, we tested the proposed method on Llama3.2-1B with two tasks selected from the GLUE benchmark: QNLI and SST-2. It is noteworthy that feature attribution for LLMs remains underexplored due to the complicated form of their outputs (sequences of tokens). To facilitate the evaluation of current feature attribution methods on LLMs, which are primarily designed for explaining classifiers, we converted Llama3.2-1B into a zero-shot classifier with task-specific prompts.
>
> - For QNLI, the prompt is as follows:
> > You are a highly accurate classifier for question-sentence pairing. Your task is to determine whether the provided sentence answers the given question. If the sentence answers the question, output 1. If the sentence does not answer the question, output 0. \
> Input:
> Question: {Input Part 1}
> Sentence: {Input Part 2}; \
> Output:
> - For SST-2, the prompt is as follows:
> > You are a sentiment classifier trained on movie reviews. Your task is to identify the sentiment of the given input. If the sentiment of the text is negative, output 0. If the sentiment is positive, output 1. \
> Input: {Input}; \
> Output:
>
> The placeholders in the task-specific prompts are later filled with content tokens from the concrete inputs. These task-specific prompts allow the concentration on the next predicted token and reduce the total number of output nodes from the vocabulary size 128,000 to 2 — corresponding to the activation of the token ‘0’ and ‘1’. Based on this setting, we deployed the explainers to determine model attributions to the content tokens. After deriving the explanations, we followed the same evaluation scheme, recording the impact of explanation-guided deletions and evaluating the effectiveness of the explanations. The experimental results are presented in the table below.
>
> | Task Type | &nbsp;&nbsp;&nbsp;&nbsp;VG&nbsp;&nbsp;&nbsp;&nbsp; | &nbsp;&nbsp;&nbsp;&nbsp;IG&nbsp;&nbsp;&nbsp;&nbsp; | KSHAP | PSHAP | &nbsp;GEFA&nbsp; | Random |
> | --- | :---: | :---: | :---: | :---: | :---: | :---: |
> | QNLI | 9.89 | 11.89 | 13.24 | 16.63 | 17.42 | 9.46 |
> | SST-2 | 9.36 | 13.98 | 15.80 | 18.62 | 19.66 | 8.30 |
>
> During the experiments on Llama, we observed that the model exhibited a neutral stance in its predictions, with the activations of the two target nodes varying within a narrow range. The limited range constrains the extent to which the model’s behavior can change, which explains the relatively smaller figures compared to the results observed in the traditional classifier settings. However, the comparison against the impacts of random removal demonstrates the effectiveness of the tested explainers in identifying relevant features. Consistent with traditional classifier settings, GEFA outperforms the selected SHAP variants, benefiting from the elimination of information waste and the application of the control variate.
>
> Another notable observation is that all black-box approaches outperform IG. However, this does not necessarily imply that black-box approaches are more suitable for explaining LLMs. Instead, we argue that these results highlight a practical challenge in applying IG: baseline selection, particularly in the domain of NLP. While black-box approaches operate in the original text space, where absence is naturally defined (i.e. replacing an absent feature with an empty token), gradient-based approaches like IG operate in the embedding space because the mapping from text space to embedding space is non-differentiable. However, the embedding space itself is typically high-dimensional and lacks intuitive interpretability, making it challenging for humans to discern its structure or the meaning of individual dimensions. Without a clear understanding of the space’s geometry, determining a suitable baseline becomes impractical, posing a significant challenge.
>
> ---
> References
>
> [1] A. Dhurandhar, K. Ramamurthy, K. Ahuja, and V. Arya. Locally invariant explanations: Towards stable and unidirectional explanations through local invariant learning. NeurIPS, 2023.
>
> [2] C. Frye, C. Rowat, and I. Feige. Asymmetric shapley values: incorporating causal knowledge into model-agnostic explainability. NeurIPS, 2020.
>
> [3] T. Heskes, E. Sijben, I. G. Bucur, and T. Claassen. Causal shapley values: Exploiting causal knowledge to explain individual predictions of complex models. NeurIPS, 2020.
>
> [4] A. Dhurandhar, K. Ramamurthy, K. Shanmugam. Is this the right neighborhood? Accurate and Query Efficient Model Agnostic Explanations. NeurIPS, 2022.
>
> [5] A. Shrotri, N. Narodytska, A. Ignatiev, J. Marques-Silva, K. S. Meel, and M. Vardi. Constraint-driven explanations of black-box {ml} models, 2021.
>
> [6] Sturmfels, Pascal, Scott Lundberg, and Su-In Lee. "Visualizing the impact of feature attribution baselines." *Distill* 5.1 (2020): e22.

---

> > ### Comment · Reviewer_ALce · 2024-11-25
> >
> > Thanks for the clarifications. Its good to see experiments with the latest llama version on some GLUE tasks although they are on mid size datasets in GLUE (MNLI and QQP are the biggest). Since ICLR allows for paper updates it would have been good to see some of the updates you mention in an updated version of the paper and how you manage to fit some of them in page limit. I feel given that your initial version still had decent amount of space on the 10th page it may not be super challenging.
> >
> > In any case, I am reasonably satisfied with the response and will raise my score to 6

---

> > > ### Author Response · Authors · 2024-11-26
> > >
> > > We appreciate the reviewer’s feedback on the additional experiments and are pleased to know that our responses have addressed the reviewer’s concerns.
> > > The exploration of the more challenging task was equally compelling to us, and we believe that the consistent performance on LLMs further supports our claim that GEFA is a general framework for deriving black-box explanations.
> > >
> > > Additionally, we would like to notify the reviewer that the revision of our manuscript is now available, with changes highlighted in blue.
> > > For better accessibility, we summarize the changes relevant to the above discussion as follows:
> > > - In Section 2: A more detailed discussion of related work in the domain of black-box approaches, including [1-5].
> > > - In Section 6: An outlook on GEFA's potential to integrate with existing approaches, which could further enhance explanation quality.

---

### Official Review · Reviewer_zURz · 2024-11-02

**Soundness:** 2
**Presentation:** 2
**Contribution:** 2
**Rating:** 3
**Confidence:** 2

**Summary:**

The authors propose a feature attribution method based on an estimator of the gradient of the model.

They propose to construct a sequence of mask and compute the gradient of the model following this path.
They construct an estimator which is the Shapley value.

**Strengths:**

The computations  described in the paper are correct.

**Weaknesses:**

Many authors have provided different ways to explain. In this paper the authors propose another one but according to me the most important work is not achieved:
1/ the authors provide an estimation on line 233 but the quality of such approximation depends on many factors, in particular on smoothness property of the evaluation of \tilde{f}. The gap between quantity is written using \approx but how the estimators coincide should deserve a proper treatment
2/ in the same way Theorem 3 provides the unbiasedness of the estimator. For me the definition is incorrect since \tilde{\xi} is not random in its construct. In the proof the authors claim that they want to prove that \xi=\tilde{\xi} but bias referes to the mean behaviour. I am quite lost in the objective of the proof.
3/ \tilde{\beta} is defined as the optimal parameter which is selected for the simulations. Yet its expression involves unknown quantity since it requires the knowledge of the distribution the way it is defined in line 802. How to compute it ?

**Questions:**

How do you select the parameters ?

---

> ### Author Response · Authors · 2024-11-23
> **Response to Reviewer zURz**
>
> While we greatly respect and value any constructive feedback, we believe this review does not adequately capture the core contributions of our work, nor does it provide any proper technical justification for the low score. Without specific, technically grounded points aligned with our core contributions, we are unable to effectively clarify or respond to the concerns raised (please refer to our detailed comments below).
>
> In fact, our paper presents a novel, self-contained framework for explainability, which is the proxy model (NOT the gradient estimation). The proposed proxy model addresses multi-model input within a unified framework, supported by axiomatic properties and rigorous derivations. This central contribution has been indeed acknowledged by the other reviewers but not at all indicated in the review at hand, rendering the absence of a proper summary. In light of these concerns, and to ensure a fair and open evaluation, we would like to inform the reviewer that we have contacted the ACs for guidance on how to address this matter. We also asked the ACs to comment on the appropriateness of this review in the context of such a high-profile conference. Our intent is to clarify any misunderstandings and improve the quality of the manuscript based on all feedback received.

---

> ### Author Response · Authors · 2024-11-23
> **Detailed comments**
>
> We sincerely appreciate the reviewer for taking the time to review our paper. We would like to take this opportunity to share our perspective on the review and address a few key concerns to foster a constructive discussion.
>
> First, we would like to clarify that the summary provided in the review does not appear to fully reflect the main contribution of the paper. The main idea of the work is NOT the gradient estimation along paths (which was in fact addressed by [1]), but rather the novel proxy model designed to address multi-modal input within a unified framework. While we are pleased that the correctness of derivations was acknowledged in the strength section, we feel that this section could have offered more specific insights into the core contributions of the paper, which would help us refine and improve the manuscript. Additionally, the statement, “*the most important work is not achieved*”, in the weakness section does not provide actionable feedback. Without clarification on what constitutes “*the most important work*” or how the paper falls short, it is difficult for us to address this concern meaningfully. We would appreciate any elaboration that could help us better understand the reviewer’s perspective.
>
> The three detailed points mentioned as weaknesses either fall outside the scope of our work or are explicitly addressed in the manuscript:
>
> - The reviewer expressed concern about the potential impact of the smoothness property of $\tilde{f}$. However, the focus of the paper is to propose a model-agnostic explanation framework, investigating the smoothness properties of specific model functions lies beyond the scope of this work. It is unclear from the context what specific discussion the comment intended to bring forward.
> - The second point relates to the unbiasedness in Theorem 3, stating “*… the definition is incorrect since $\tilde{\boldsymbol{\xi}}$ is not random in its construct*”. The randomness of $\tilde{\boldsymbol{\xi}}$ is introduced through mask sampling, as acknowledged in the summary section of the review itself: “*They propose to construct a sequence of **mask** …*”.
> - The same point also questions the definition of unbiasedness, stating “*…  bias referes to the mean behaviour*”. We would like to point out that $\tilde{\boldsymbol{\xi}}$ is built upon gradient estimators, as noted again in the summary section of the review: “They construct an **estimator …**", and the term “estimator” inherently refers to the mean behavior.
> - The last point is about the computation of $\beta^*$, which is explicitly addressed in lines 281-283 in the original manuscript. We state that the unknown quantity $\mathrm{Cov}(f, h)$ can be estimated through collected observations and cite the reference [2] for the validity of the solution.
>
> Regarding the parameter selection raised in the question section, we empirically set the query budgets for the black-box approaches to 500 and 5,000 for the text and image classifiers, respectively. The explainers receive a larger budget for the image classifier due to the higher dimensionality of its input feature space. Across the different settings of the experiment, the query budgets for the black-box approaches, including GEFA, were always set to the same value to ensure a fair comparison. For the baseline choice, we followed the suggestion by [3] for explaining the image classifier and adopted the empty token for the text classifier. Specifically, replacing a token with an empty token removes the corresponding feature when it is switched off during sampling. Using an empty token as the baseline aligns with the natural definition of absence and is only possible for text classifiers, which can handle inputs of varying lengths. Lastly, while $\beta$ might appear to be a parameter, we would like to clarify that its value is computed automatically based on the collected observations. As such, the value of $\beta$ does not require manual specification.
>
> We sincerely hope this follow-up provides additional context to support the reviewer’s evaluation, and we are happy to address any further questions or concerns that the reviewer might have.
>
> ---
> References
>
> [1] Cai, Yi and Gerhard Wunder. "On gradient-like explanation under a black-box setting: When black-box explanations become as good as white-box." In International Conference on Machine Learning, pp. 5360–5382. PMLR, 2024.
>
> [2] Mohamed, Shakir, et al. "Monte Carlo gradient estimation in machine learning." Journal of Machine Learning Research 21.132 (2020): 1-62.
>
> [3] Sturmfels, Pascal, Scott Lundberg, and Su-In Lee. "Visualizing the impact of feature attribution baselines." Distill 5.1 (2020): e22.

---

> ### Author Response · Authors · 2024-11-26
>
> We would like to thank the reviewer once again for taking the time to review our paper. We remain open to any follow-up discussions that could help clarify the contributions and insights presented in the manuscript, and we look forward to the feedback and any additional thoughts the reviewer may wish to share.

---

> ### Author Response · Authors · 2024-12-02
>
> Dear Reviewer,
>
> As the extended rebuttal phase concludes today, we would like to follow up on our responses to your comments and inquire if they sufficiently address your concerns.
> Should you have any remaining concerns or require further clarification, we would be happy to address them promptly.
>
> Thanks again for your time and feedback throughout this process.

---

### Official Review · Reviewer_QMVW · 2024-11-03

**Soundness:** 4
**Presentation:** 3
**Contribution:** 4
**Rating:** 8
**Confidence:** 3

**Summary:**

This paper presents GEFA (Gradient-estimation-based Explanation For All), which extends previous work on gradient estimation-based model explanations to handle both discrete and continuous features. The key innovation is introducing proxy variables that represent feature presence probabilities, allowing the method to work with any input type. It is proven that GEFA produces unbiased Shapley value estimates and introduce a variance reduction technique using control variates. The method is evaluated on both text and image classification tasks.

**Strengths:**

I think the paper's strengths are best contextualized as a follow up on GEEX. GEEX focused on continuous features and image classifiers, using gradient estimation in the original feature space with Gaussian noise perturbations. It proved alignment with Integrated Gradients but lacked variance reduction.

GEFA handles both discrete and continuous features through proxy variables representing feature presence probabilities. It uses binary mask sampling, proves unbiased Shapley Value calculation, and introduces feature correlation-based variance reduction. By operating in a proxy space rather than the original feature space, GEFA overcomes GEEX's continuous-only limitation. While GEEX was tested only on images, GEFA demonstrates broader applicability through evaluations on both image tasks (InceptionV3 on ImageNet) and text data (BERT on Amazon reviews), effectively bridging gradient-based and Shapley approaches.

**Weaknesses:**

- I think the analysis could benefit from more diverse text tasks beyond sentiment analysis, and on tabular datasets.
- A thorough review of computational complexity would be appreciated.
- What happens when the assumption about feature presence correlation is violated?
- Why this proxy path? This could use more detail.
- Optimal beta values assume feature independence which usually does not hold in the real world.

**Questions:**

- How sensitive is the method to baseline image? Why use blurred images, instead of black? What about other domains?
- Do you think the control variate assumption about correlation between number of present features and model outputs is justified?
- Have you explored other proxy paths beyond the straightline? What are the tradeoffs?
- Could we apply the variance technique on other black box explanation methods?
- Is there a good way to detect when the feature presence correlation is violated?

---

> ### Author Response · Authors · 2024-11-23
> **Response to Reviewer QMVW**
>
> We would like to thank the reviewer for acknowledging our efforts in developing a novel framework for deriving black-box explanations and for providing constructive comments. As the reviewer raises multiple points, we address each of them individually below.
>
> **Extending experiments**: We fully agree that GEFA can be applied to broader settings. For this point, we would like to refer the reviewer to our discussion with Reviewer ALce, where we reported additional experiments on Llama3.2, with two tasks selected from the GLUE benchmark.
>
> **Computational complexity**: The computational complexity of black-box explanation methods is primarily composed of two factors: query generation and model inference. Given a total budget of $n$ queries in a $p$-dimensional feature space, the cost of query generation for GEFA is $\mathcal{O}(np)$. Denoting the cost of model inference by $\mathcal{O}(\mathcal{M})$, the overall complexity of GEFA becomes $\mathcal{O}(np+n\mathcal{M}))$, which is equivalent to the complexity of GEEX. While both factors contribute to the complexity of GEFA, the total time cost in practice is dominated by $\mathcal{O}(n\mathcal{M})$, as GEFA allows for the pre-generation of query masks at its initialization. The pre-generated masks significantly reduce the complexity associated with query generation. For PSHAP and KSHAP, the complexity increases to $\mathcal{O}((n+\epsilon)\cdot p+n\mathcal{M})$, where the additional cost $\epsilon$ arises from partitioning the feature space (PSHAP) or solving linear regression (KSHAP). In our experiments, the query budget was fixed, so we did not report the time cost for deriving each explanation. However, the recorded time costs per instance (presented in the table below) align with the above analysis.
>
> | Time cost (s) | &nbsp;&nbsp;&nbsp;&nbsp; IG &nbsp;&nbsp;&nbsp;&nbsp; | KSHAP | PSHAP | &nbsp;GEEX&nbsp; | &nbsp;GEFA&nbsp; |
> | --- | :---: | :---: | :---: | :---: | :---: |
> | BERT | 0.64 | 1.71 | 1.55 | - | 1.38 |
> | InceptionV3 | 1.35 | - | 15.63 | 4.57 | 4.91 |
>
> **Validity of the correlation assumption**: We believe that the assumption regarding feature presence correlation holds in most cases, particularly when viewed from a probabilistic perspective. Consider a model that learns to use a subset of features for a specific decision, the presence of these relevant features triggers stronger activations in the model’s output. When more features are present, the probability of triggering a strong activation increases, and vice versa. In other words, a higher ratio of feature presence is expected to correlate with stronger model activations in expectation, thereby supporting the correlation assumption.
>
> **Violation of correlation assumption**: We understand the concern regarding potential violations of the correlation assumption, though such cases are rarely observed in practice. However, even in scenarios where this assumption does not hold, the issue is automatically addressed by the weighting parameter $\beta^*$, which is computed as $\beta^*=\mathrm{Cov}(f,h)/\mathrm{Var}(h)$. If there is no correlation between the model outcome $f(\cdot)$ and feature presence $h(\cdot)$, the covariance term $\mathrm{Cov}(f,h)$ becomes zero, causing $\beta^*$ collapse to 0. The zero-valued $\beta^*$ eliminates any potential impact introduced by the control variate. The computation of $\beta^*$ inherently serves as a detector for the validity of the correlation assumption, which should address the last question in the reviewer’s comment.
>
> **Optimal beta and feature independence**: We would like to clarify that the optimal beta value does not rely on the assumption of feature independence. Instead, it assumes that each individual feature has a relatively minor contribution to the prediction function, especially in high-dimensional feature spaces. Although this assumption is not strictly valid, it serves to significantly simplify the process of obtaining a pseudo-optimal value for $\beta$.
>
> **Why the straightline path**: The motivation for choosing the straightline path is to satisfy the property of symmetry [1]. Deviating from the diagonal of the proxy space will violate this property. Furthermore, our analysis of the relationship between GEFA and IG (Section 4.4 and Appendix A.4) concludes that the straightline path in the proxy is equivalent to the average of all $p!$ unique edge paths. This equivalence allows for transforming the task of averaging estimates across multiple paths into the simpler task of estimating attributions along the diagonal of the proxy space.

---

> ### Author Response · Authors · 2024-11-23
> **Continued response to detailed points**
>
> **Baseline choice**: The baseline choice is a crucial topic for feature attribution methods, as it largely affects the explanation results. In this work, we use blurred images as the baseline for image data following the suggestion by [2], which conducted a thorough study on the influence of baseline choice. For textual data, we define the baseline token as an empty token, aligning with the natural definition of absence. Replacing a token with an empty token removes the corresponding feature when it is switched off during sampling. However, it is noteworthy that there is yet no “golden rule” for baseline selection given the variety of modalities. Most baseline choices in literature are guided by intuition and specific use cases, highlighting the need for further investigation.
>
> **Applicability of variance reduction**: Yes, we think the control variate technique is applicable to other black-box approaches, which can be viewed as estimators for some underlying attribution values. However, this exact form of the control variate $h(\cdot)$ should be re-designed for each explainer to ensure that the property of unbiasedness is preserved.
>
> ---
> References
>
> [1] Sundararajan, Mukund, Ankur Taly, and Qiqi Yan. "Axiomatic attribution for deep networks." *International conference on machine learning*. PMLR, 2017.
>
> [2] Sturmfels, Pascal, Scott Lundberg, and Su-In Lee. "Visualizing the impact of feature attribution baselines." *Distill* 5.1 (2020): e22.

---

> > ### Comment · Reviewer_QMVW · 2024-11-30
> >
> > Thank you for your note. I think this work is important and technically sound, so I will keep my rating of 8.

---

### Official Review · Reviewer_UNFo · 2024-11-08

**Soundness:** 3
**Presentation:** 3
**Contribution:** 3
**Rating:** 5
**Confidence:** 3

**Summary:**

The paper derives a black-box feature attribution method that only asks for queries. The key idea is to state feature importance in terms of probability of presence of each feature. This turns out to produce an equivalent of Shapley values.

**Strengths:**

- Interesting way to generalize "gradients"
- clean derivation and presentation
- sound results and theoretical connections
- Important deviation from noising-based estimation of attribution.

**Weaknesses:**

The metric of choice is not fully explained. I get that the removing supposedly relevant inputs should lead to larger AOPC. However, the model that is being used to predict the label needs to be retrained to ensure that the input  has not gone output of support to what the model was trained on. Hooker et al. point this out.

Both https://arxiv.org/abs/1806.10758, https://arxiv.org/abs/2103.01890 point this out. Am I missing that the model for the scoring was retrained or trained in a special way? I'm very skeptical of what the results say without fixing this issue. Further, there is also the problem of encoding which is mentioned in the second paper (that is formally shown to make removal based evaluations improper, https://arxiv.org/abs/2411.02664). The first two papers need to be discussed. The latter is more of a reference.

Can the authors say more about how their APOC evaluation is sufficient? If not, is there a possibility to retrain the scorer model and recompute AOPC? Otherwise, I am not sure how much to trust the numbers in the tables.

**Questions:**

See weaknesses.

---

> ### Author Response · Authors · 2024-11-23
> **Response to Reviewer UNFo**
>
> We would like to thank the reviewer for acknowledging our efforts in developing a novel framework for deriving black-box explanations.
> We also appreciate the detailed comments and the thought-provoking discussion on explanation evaluation, accompanied by a valuable list of references, which we found to be very insightful.
>
> Indeed, compared to the development of explanation methods, the evaluation of explanation quality remains an open challenge, with no established consensus in the field. In this work, we adopted the evaluation via deletion approach, a commonly used method for accessing explanation quality. The employment of this approach spans from the early stages of explainability research [1, 2] to the most recent studies[3, 4]. One of its key advantages is that it does not require retraining during evaluation, thereby offering a static environment for efficient explanation quality assessment.
>
> Regarding the out-of-distribution issue, we would like to emphasize that the performance of random feature removal — used as a baseline competitor (referred to as the control variant by Hooker [5]) — alleviates this concern. The significantly lower nAOPC scores of random removal demonstrate that simply shifting the input away from the underlying data manifold does not effectively affect model performance. Furthermore, our findings on the extended experiment (which will be discussed later) suggest that nAOPC acquired through evaluation via deletion yields results comparable to those obtained using retraining-based schemes, supporting its validity as an evaluation method.
>
> **TL; DR**: The remove and retrain (ROAR) scheme yields different observations due to the presence of residual features with negative attributions, which preserve task-relevant information and lead to higher accuracies when retraining on the manipulated dataset. We suggest that “keep and retrain” (KEAR) avoids this issue, offering a more faithful quantification of explanation quality.
>
> Specifically, we would like to begin by discussing [5] and justifying the claim outlined above. An important conclusion drawn by [5] is that many attribution methods “*are not better than a random designation of feature importance*” according to results from the ROAR scheme, which highlights their concerns regarding the traditional deletion metric. However, this conclusion appears to contradict the widely approved effectiveness of IG. The apparent conflict motivated our further investigation, which revealed **that the presence of residual features with negative attributions is the root cause of the misalignment between observations from different evaluation perspectives**.
>
> **The “sign” issue**: In the context of feature attribution, a positive attribution score indicates a positive contribution to the prediction result, whereas a negative score, rather than indicating irrelevance, represents a negative association with the decision. Failing to remove negatively contributing features preserves task-relevant information, which the model can reorganize during retraining to improve accuracy. A qualitative example of residual “negative” information is illustrated in the visualized pixel removal process of IG in Figure 1 of [5]. This “sign” issue also explains the effective manipulation by SG-SQ and VarGrad, which provide unsigned attributions, thereby ensuring the removal of all informative features.
>
> To mitigate the assessment distortion caused by retained negative information, we suggest a shift in methodology: rather than removing features for retraining, top-ranked features should be retained. The “keep and retrain” approach reframes the evaluation question as:
>
> > “Does an explanation method effectively identify relevant features?”
>
> An effective explanation method should capture relevant information, resulting in higher accuracy for the retrained model with the same portion of retained information. This exact experiment has been conducted by [5] and reported in Table 2 on page 17. The first 4 columns of the table (related to “keep and retrain”) demonstrate that IG is among the best-performing explanation methods, as evidenced by the high accuracies of the retrained model. This observation complies with the widely recognized effectiveness of IG in the literature based on the traditional deletion scheme.
> To explore this further, we plan to contact the authors of [5] to continue the discussion on this matter.

---

> ### Author Response · Authors · 2024-11-23
> **Experimental results following ROAR scheme**
>
> **Our results on ROAR and KEAR**: Although we argue that the nAOPC reported on the traditional deletion scheme provides a faithful reflection of explainer performance, we agree that extending the evaluation to include retraining schemes enhances the persuasiveness of the results. To address the reviewer's suggestion, we conducted additional experiments following the ROAR framework. Specifically, we fine-tuned EfficientNet-B0 on the Cats vs Dogs dataset and created copies of the dataset with explanation-guided manipulation. These manipulated datasets were then used for retraining the pre-trained model to assess the quality of explanations.
>
> Due to time constraints, we downsampled the dataset to 2000/400/400 partitions for training, validation, and test sets. The downsampled dataset enabled EfficientNet-B0 to achieve an accuracy of 99.40% after fine-tuning. Based on the fine-tuned EfficientNet-B0, we generate explanations for images in all three partitions using three competitors: IG, PSHAP, and GEFA. Features for each instance were ranked in descending order according to their attribution scores.
>
> The top 90% of features were **removed** for the ROAR test, whereas the top 10% of features were **kept** for the KEAR test. To highlight the “sign” issue, we also performed feature ranking based on the absolute values of their attribution scores for the removal test, referred to as ROAR-abs (remove and retrain — based on absolute attribution score). The results from these three settings are summarized in the following table. Please note that lower retraining accuracy indicates better explanation quality in the removal tests (ROAR and ROAR-abs), whereas higher accuracy reflects superior explainer performance in KEAR.
>
> | Acc. | &nbsp;&nbsp;&nbsp;&nbsp;IG&nbsp;&nbsp;&nbsp;&nbsp; | PSHAP | &nbsp;GEFA&nbsp; | Random |
> | --- | :---: | :---: | :---: | :---: |
> | ROAR 90% | 77.20 | 79.25 | 82.35 | 71.30 |
> | KEAR 10% | 89.60 | 84.30 | 89.95 | 71.30 |
> | ROAR-abs 90% | 62.80 | 76.75 | 68.45 | 71.30 |
>
> In the ROAR 90% test, all explanation methods show minor manipulation impacts due to the previously discussed “sign” issue and fail to excel random removal. By addressing the distortion caused by residual information, the KEAR 10% test offers a more faithful assessment of explanation quality. The success of the explainers in identifying the most informative features results in relatively high classification accuracy with only 10% of features retained. Notably, these findings align closely with the observations in Table 2 of [5]. The last row — ROAR-abs 90% — demonstrates that removing residual negative information enhances the effectiveness of manipulation, serving as indirect evidence for the “sign” issue. However, the origin of negative attributions is complex, influenced by various factors such as the baseline choice and a feature’s association with the prediction function. Ranking features based on the absolute value of attribution scores can unnecessarily include irrelevant features, as it cannot distinguish between different causes of a negative sign. Consequently, this approach remains a suboptimal solution compared to KEAR.
>
> Lastly, we present our nAOPC results obtained following the traditional deletion approach, alongside the KEAR results, to show that the retraining scheme and recursive deletion scheme are parallel evaluation options rather than contradictory methods. As the metrics employed in the two schemes differ in scale, the numerical values across settings are not directly comparable. However, the consistency in relative rankings within each test provides a meaningful reference for the validity of nAOPC.
>
> |  | &nbsp;&nbsp;&nbsp;&nbsp;IG&nbsp;&nbsp;&nbsp;&nbsp; | PSHAP | &nbsp;GEFA&nbsp; | Random |
> | --- | :---: | :---: | :---: | :---: |
> | KEAR 10% (in Acc.) | 89.60 | 84.30 | 89.95 | 71.30 |
> | nAOPC | 40.82 | 39.56 | 40.79 | 35.07 |

---

> ### Author Response · Authors · 2024-11-23
> **Discussion on the encoding issue**
>
> The reviewer also suggested [6, 7] regarding the **encoding issue**, which are solid work and interesting readings for us. The encoding issue is a common challenge in instance-wise feature selection (IWSF). However, we believe it is not a significant concern for feature attribution methods in general. While IWSF and feature attribution are closely related, they provide explanations from fundamentally different perspectives.
>
> IWSF provides **data-oriented explanations**, aiming to identify the most relevant features to a task based on instance-label pairs observed in a dataset. We use the term “data-oriented” since IWSF determines feature importance directly from the information presented by the dataset. Determining feature importance according to the given dataset makes IWSF particularly useful in the upstream stages of the machine learning pipeline, contributing to tasks like feature selection (as the name indicates), narrowing the feature space, and promoting the efficiency of model training.
>
> In contrast, feature attribution studied in this paper provides **model-oriented explanations**, focusing on how a model processes input data. Feature attribution methods investigate the extent to which each feature contributes to a specific decision, and their outputs depend entirely on the behavior of the target model. Since feature attribution is not concerned with dataset-level patterns or training processes, the encoding issue is a less significant concern for this category of approaches, whose focus is fundamentally different.
>
> ---
> References mentioned in the comments:
>
> [1] Samek, Wojciech, et al. "Evaluating the visualization of what a deep neural network has learned." IEEE transactions on neural networks and learning systems 28.11 (2016): 2660-2673.
>
> [2] Montavon, Grégoire, Wojciech Samek, and Klaus-Robert Müller. "Methods for interpreting and understanding deep neural networks." Digital signal processing 73 (2018): 1-15.
>
> [3] Muzellec, Sabine, et al. "Saliency strikes back: How filtering out high frequencies improves white-box explanations." In International Conference on Machine Learning, pp. 5360–5382. PMLR, 2024.
>
> [4] Cai, Yi and Gerhard Wunder. "On gradient-like explanation under a black-box setting: When black-box explanations become as good as white-box." In International Conference on Machine Learning, pp. 5360–5382. PMLR, 2024.
>
> [5] Hooker, Sara, et al. "A benchmark for interpretability methods in deep neural networks." The Thirty-third Annual Conference on Neural Information Processing Systems.
>
> [6] Jethani, Neil, et al. "Have We Learned to Explain?: How Interpretability Methods Can Learn to Encode Predictions in their Interpretations." International Conference on Artificial Intelligence and Statistics. PMLR, 2021.
>
> [7] Puli, Aahlad Manas, Nhi Nguyen, and Rajesh Ranganath. "Explanations that reveal all through the deﬁnition of encoding." The Thirty-eighth Annual Conference on Neural Information Processing Systems.

---

> ### Author Response · Authors · 2024-11-26
>
> We would like to thank the reviewer again for raising the discussion on explanation evaluation.
> We are writing to notify the reviewer that a revision of our manuscript is now available, which includes our analysis and experiments on the retraining scheme to address the concerns raised.
>
> For better accessibility, we summarize the changes relevant to the above discussion as follows:
> - In Section 5.1: We refer to [1] and [2] regarding out-of-distribution concerns and evaluation validity, and include a clear instruction directing readers to Appendix B for additional discussions.
> - Appendix B: A newly added section that consolidates our responses on the validity of the adopted evaluation scheme. This section includes:
>     - An analysis of the misalignment between different evaluation perspectives and a justification for the modification to the retraining scheme;
>     - Experiments conducted under the retraining scheme and interpretations of the results.
>
> We hope these revisions address the reviewer’s concerns, and we would greatly appreciate any further feedback.
>
> [1] Hooker, Sara, et al. "A benchmark for interpretability methods in deep neural networks." The Thirty-third Annual Conference on Neural Information Processing Systems.
>
> [2] Jethani, Neil, et al. "Have We Learned to Explain?: How Interpretability Methods Can Learn to Encode Predictions in their Interpretations." International Conference on Artificial Intelligence and Statistics. PMLR, 2021.

---

> ### Author Response · Authors · 2024-12-02
>
> Dear Reviewer,
>
> As the extended rebuttal phase concludes today, we would like to follow up on our responses to your comments and inquire if they sufficiently address your concerns.
> Should you have any remaining concerns or require further clarification, we would be happy to address them promptly.
>
> Thanks again for your time and feedback throughout this process.

---

### Author Response · Authors · 2024-11-25
**Reminder: Rebuttal Phase Ending Soon**

Dear Reviewers,

We would like to kindly remind you that the rebuttal phase soon approaches its end.

Should there be any remaining concerns or clarifications needed, we would be happy to address them promptly.

Thanks again for your time and your valuable feedback throughout this process.

---

### Meta-Review · Area_Chair_CVZH · 2024-12-22

**Metareview:**

This paper develops gradient based explanations for black box models by using score function gradients to produce gradients of the expectation of the model where inputs are randomly masked out inputs. These gradients are then integrated at different probabilities to form the explanation GEFA. GEFA was proved to be a Shapley value and there's a discussion on how to reduce the variance of score function gradients. The reviewers were mixed. Concerns about related work and evaluations were raised. Some of these were addressed in the update and the author rebuttal.

The authors considered ROAR, which was seen as a strength by the reviewers. There was also a discussion about the merits of ROAR in the author response. I think some of that should be more heavily worked into the main paper. Similarly, there was a discussion about encoding in the author replies and whether it was relevant as one of the papers provided by a reviewer on encoding proves negative results about ROAR. That discussion was unclear to me as the model and the data are the same and can provide "samples" from their respective distributions, so whether encoding is a problem or not depends more than on just whether the question is about models versus data. This needs some clarification in the main paper as well.

Overall, I felt the work was interesting and the reviewers comments improved the paper. However, one more round of revision by the authors will really strengthen the work.

**Additional Comments On Reviewer Discussion:**

See the metareview for the main discussion. The concerns about evaluations and the relatively lengthy author replies swayed my decision that the work would benefit from one more round of polish,.

---

### Decision · Program_Chairs · 2025-01-22

Reject